# The configuration of Northern Hemisphere ice sheets through the Quaternary

Christine L. Batchelor[1,2], Martin Margold[3], Mario Krapp[4], Della K. Murton[4], April S. Dalton [5], Philip L. Gibbard[1], Chris R. Stokes [5], Julian B. Murton[6] & Andrea Manica [4]

Our understanding of how global climatic changes are translated into ice-sheet fluctuations and sea-level change is currently limited by a lack of knowledge of the configuration of ice sheets prior to the Last Glacial Maximum (LGM). Here, we compile a synthesis of empirical data and numerical modelling results related to pre-LGM ice sheets to produce new hypotheses regarding their extent in the Northern Hemisphere (NH) at 17 time-slices that span the Quaternary. Our reconstructions illustrate pronounced ice-sheet asymmetry within the last glacial cycle and significant variations in ice-marginal positions between older glacial cycles. We find support for a significant reduction in the extent of the Laurentide Ice Sheet (LIS) during MIS 3, implying that global sea levels may have been 30–40 m higher than most previous estimates. Our ice-sheet reconstructions illustrate the current state-of-the-art knowledge of pre-LGM ice sheets and provide a conceptual framework to interpret NH landscape evolution.

[1] Department of Geography, University of Cambridge, Scott Polar Research Institute, CB2 1ER Cambridge, UK. [2] Department of Geoscience and Petroleum, Norwegian University of Science and Technology (NTNU), NO-7491 Trondheim, Norway. [3] Department of Physical Geography and Geoecology, Charles University, 128 43 Prague, Czech Republic. [4] Department of Zoology, University of Cambridge, CB2 3EJ Cambridge, UK. [5] Department of Geography, Durham University, DH1 3LE Durham, UK. [6] Department of Geography, University of Sussex, BN1 9RH Brighton, UK. Correspondence and requests for materials should be addressed to C.L.B. (email: clb70@cam.ac.uk)

The growth and decay of continental ice sheets have formed an integral part of the Earth's climate system during the Late Cenozoic and particularly over the last 2.6 Ma (the Quaternary Period), resulting in major fluctuations in global sea level[1]. Accurately reconstructing the former extent of ice sheets is, therefore, vital to understand how global climatic changes are translated into ice-sheet fluctuations, providing important constraints for future predictions of sea-level change[2]. Furthermore, knowledge of the configuration and evolution of palaeo-ice sheets through time is required to understand their impact on a wide range of important issues across numerous disciplines, including the Earth's rheology, long-term landscape evolution[3], palaeoecology[4], genetic diversity[5] and anthropology[6]. Over the last few decades, unprecedented growth in the size and diversity of empirical datasets used to reconstruct the extent of palaeo-ice sheets, together with major improvements in our ability to numerically model their dynamics[7], have led to important advances in our understanding of ice-sheet configuration through time. However, the vast majority of these reconstructions[8–12] focus on ice-sheet deglaciation since the Last Glacial Maximum (LGM) c. 26.5 ka[13]. In contrast, there have been few attempts at constraining the extent of ice sheets prior to the LGM[14,15]. This is largely because of the paucity of empirical data, which are highly fragmentary in both space and time[16], and has led to an over-reliance on loosely constrained and/or coarse-resolution numerical modelling at the global or hemispheric scale[17–20]. Thus, we have very limited knowledge of the Earth-surface conditions of the mid- and high-latitudes throughout most of the Quaternary.

To address this issue, we take a consistent methodological approach in synthesising empirical data and numerical modelling results related to pre-LGM ice sheets to produce testable hypotheses of Northern Hemisphere (NH) ice-sheet configurations at key time-slices spanning the Quaternary. These hypothesised ice-sheet extents are used to assess spatial differences in ice-sheet configuration within and between glacial periods, produce new first-order estimates of global sea level associated with each time-slice, and explore the implications for long-term landscape evolution.

## Results

**Reconstruction of ice-sheet extents.** Empirical evidence relating to NH ice sheets, together with the output from numerical models, from over 180 published studies is compiled for 17 pre-LGM time-slices that extend back to the Late Pliocene (Fig. 1, Supplementary Figures 1–10, Supplementary Tables 1–17). Although ice sheets also fluctuated in the Southern Hemisphere (in Antarctica, Patagonia and New Zealand), the major mid-latitude ice sheets of the NH dominated fluctuations in the global sea-level record[21]. In this study, maps showing the available evidence relating to past ice-sheet extent (e.g. Figure 1a) are produced at 5 ka intervals during ice-sheet build-up prior to the LGM, for MIS 4 and 5a–d, and for a further six major glaciations extending back to MIS 20–24 (790–928 ka)[22] (Supplementary Figures 2–9). Terrestrial evidence for glaciations older than 1 Ma, during the Early Pleistocene to Late Pliocene, is scarce and dated mostly by palaeo-magnetic methods[23,24]. These intervals are therefore grouped into two broad time-slices: the early Matuyama magnetic Chron (1.78–2.6 Ma), which encompasses the onset of major NH glaciation recorded by terrestrial evidence, around 2.4–2.5 Ma;[14,25,26] and the late Gauss Chron (2.6–3.6 Ma), which includes the onset of major NH glaciation recorded by ice-rafted debris in ocean cores, around 2.6–2.7 Ma[27,28] (Fig. 1c, Supplementary Figures 9 and 10). Our maps of evidence relating to pre-LGM ice sheets (e.g. Figure 1a) reveal the geographical regions and time-slices in which empirical data are sparse and/or conflicting (Supplementary Figures 2–10).

Empirically derived and numerically modelled outlines of ice-sheet extent were the primary targets of our literature search for evidence for NH ice sheets. Although it is beyond the scope of this study to review all marine-sedimentological evidence for ice-sheet growth and decay (e.g. ice-rafted debris), evidence derived from sedimentological and stratigraphic investigations was incorporated into our reconstructions (Supplementary Tables 1–17). These data types were specifically targeted for older time-slices for which published ice-sheet outlines are scarce. With the exception of the comparatively warm periods of 45 ka, MIS 5a and 5c, for which we aim to capture the ice-sheet configurations during peak warmth, our reconstructions aim to show the maximum ice-sheet extent within each time-slice (Methods). This is particularly important to note for the oldest time-slices (i.e. early Matuyama and late Gauss magnetic chrons), which span long periods of time that included significant fluctuations in ice-sheet extent[29].

Following the compilation of the available evidence, we then produce new hypotheses relating to ice-sheet extent that span the Quaternary (Fig. 1d–u). For each time-slice we capture uncertainty by defining a maximum and minimum limit allowed by the available evidence (Fig. 1a, b) and provide a best-estimate hypothesis (Fig. 1d–u, Supplementary Figures 2–10). Max–min bounds have been used previously to illustrate uncertainty in the past extent of ice masses[11]. Our best-estimate reconstructions are scored from low to high confidence using a robustness score (Fig. 1d–u) that is based on the availability and agreement between various modelled and empirical constraints for that time-slice. Some of our reconstructions are well constrained by empirical data, especially for more recent time-slices, e.g. the maximum extent of the NH ice sheets during MIS 6 is generally very well constrained (Fig. 1a, b). However, comparatively few data about ice-sheet extent exist during older time-slices, interstadial periods (e.g. 45 ka, MIS 5a and 5c), and glacial periods such as MIS 8 and 10 that occurred between glaciations of greater extent. There is also spatial variability in the distribution of empirical data, with information about past ice sheets particularly limited from north-east (NE) Asia (Supplementary Figures 2–10).

In regions where there are few or no existing data for a time-slice, we use a reconstruction from another time-slice that has a similar value in the benthic $\delta^{18}O$ stack[1] to construct a plausible ice-sheet margin (Methods, Supplementary Notes 1–18). Thus, some of our older reconstructions are based, in part, on ice-sheet extents from younger time-slices. For example, the best-estimate LIS during MIS 12 incorporates the best-estimate reconstruction for MIS 6 where empirical data[26] are absent (Supplementary Note 14). To avoid unnecessary complexity in regions where empirically derived reconstructions are scarce, ice-sheet templates were used for the North American Cordillera, Greenland, Iceland and NE Asia (Methods). For example, three ice-mass configurations are used for NE Asia: the Pleistocene maximum[30,31], the LGM[31], and no ice sheet. The use of templates and ice-sheet extents from other time-slices is necessary to fill the gaps in our current knowledge of Quaternary ice-sheet extent, and is an improvement on methods that use the LGM as input for all Quaternary glaciations.

In total, we reconstruct a maximum, minimum and best-estimate NH ice-sheet extent for 17 separate time-slices prior to the LGM, and a best-estimate for the comparatively well-constrained LGM[8,11,13,14]. Although our best-estimate ice-sheet reconstructions are informed by some subjective decisions, they provide the first set of consistently generated reconstructions of NH ice sheets through the Quaternary that are based on available

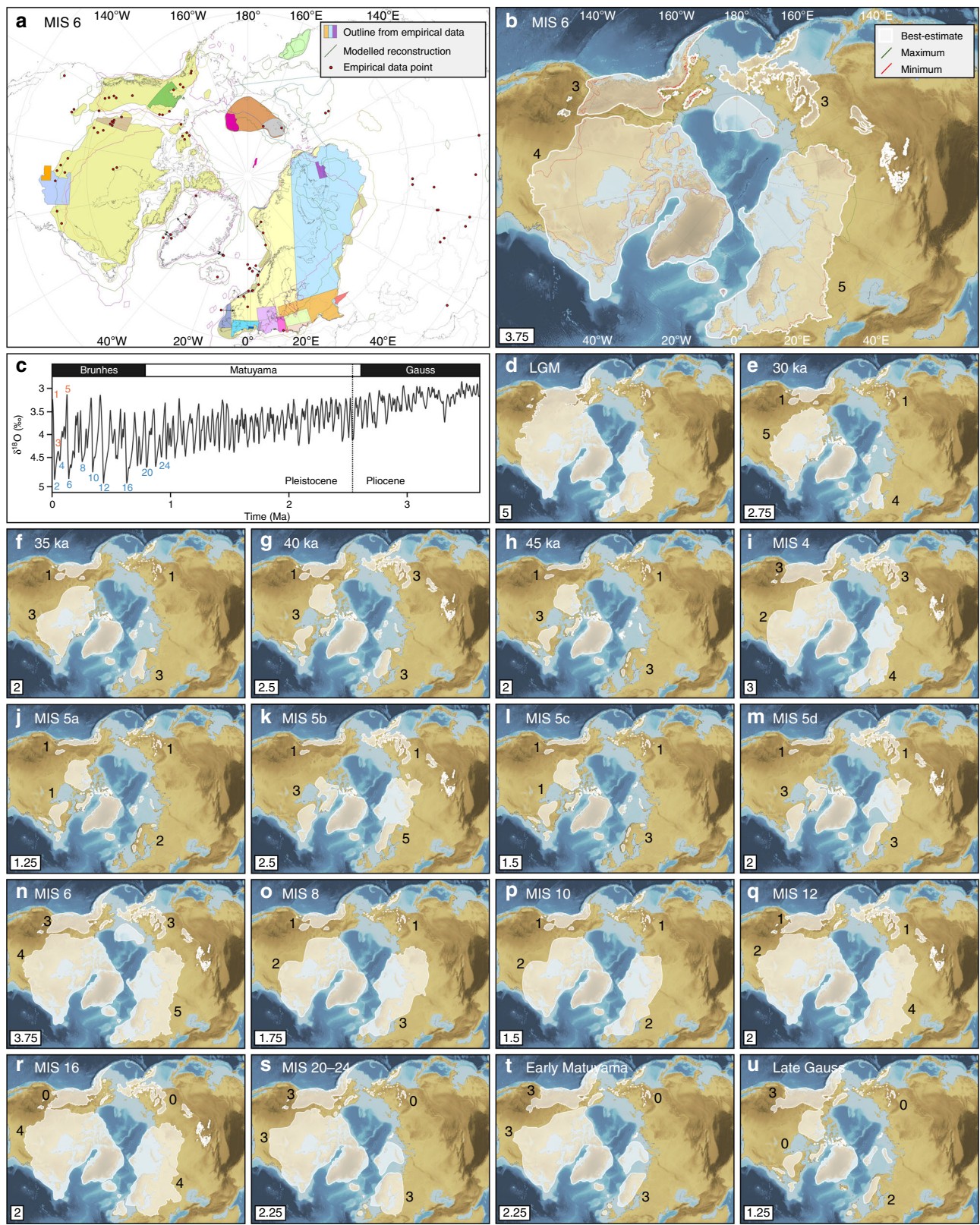

empirical evidence. Note that whereas our ice-sheet reconstructions for the last glacial cycle (MIS 2–5d) represent the likely chronological maximum extent, the mapping of time-transgressive ice margins for time-slices older than the last glacial cycle is precluded by the fragmentary nature of the empirical data and problems of dating older glacial sediments at sub-stage resolution.

**Variations in ice-sheet extent**. Our reconstructions clearly illustrate spatial differences in the configuration of NH ice sheets in different glacial cycles since the Late Pliocene (Figs. 1d–u and 2). During the most recent and best-constrained glacial cycle (MIS 2–5d; Fig. 1d–m), our detailed reconstructions of ice-sheet chronological extent support the hypothesis[9] that glaciers and ice sheets developed in continental interiors (i.e. NE Asia and eastern

**Fig. 1** Hypothesised reconstructions of NH ice-sheet extent during the Quaternary. **a** shows how data sources are compiled for the example time-slice of MIS 6 (132–190 ka). Data key is in Supplementary Table 10. **b** shows maximum, minimum and best-estimate reconstructions of ice-sheet extent during MIS 6, which are derived from the data in **a**. The decisions made in producing these reconstructions are explained in Supplementary Note 11. **c** is the benthic $\delta^{18}O$ stack for the Pleistocene and Late Pliocene[1]. Blue and orange numbers show the marine isotope stages (MIS) corresponding to cool and warm periods, respectively, for which reconstructions of ice-sheet extent are produced in this study. **d–u** are best-estimate reconstructions of NH ice-sheet extent for 18 time-slices through the Quaternary. The overall robustness score (Methods) for each time-slice reconstruction is shown in the bottom left corner. Black numbers are individual ice-sheet robustness scores. Background is ETOPO1 1 arc-minute global relief model of the Earth's surface (https://www.ngdc.noaa.gov/mgg/global/)[72]. Large versions of all maps are available in Supplementary Figures 2–10

Europe) early in the last glacial cycle, whilst large ice sheets close to maritime moisture sources (i.e. western European Ice Sheet (EIS) and Laurentide Ice Sheet (LIS)) attained their maximum extent towards the end of the glacial cycle. A comparison between the LGM and MIS 4 ice extents (Fig. 3a), for example, shows that the southern and western margins of the LIS and the EIS were more extensive during the LGM, whereas the eastern margin of the EIS and glaciation in NE Asia and the North American Cordillera were more extensive during MIS 4. Ice sheets in eastern Europe and NE Asia were probably of similar size or even more extensive in MIS 5b and/or 5d compared to MIS 4[32] (Fig. 1i, k and m). These spatial patterns in Late Pleistocene ice extent suggest that glaciation may be initiated in the Pacific region, before spreading to the North Atlantic region. Although it is not currently possible to assess geological evidence for NH ice-sheet asynchronicity within older glacial periods, records of global dust flux derived from Antarctic ice cores show a pronounced double peak within many earlier glacial cycles[15], suggesting that a two-stage pattern of ice-sheet development may also have occurred during older glaciations.

The asynchronous development of the NH ice sheets has been attributed to ice-sheet growth causing an increase in global aridity through each glacial cycle, with large ice sheets close to maritime moisture sources being less sensitive to a reduction in moisture supply[9,15]. The extent and elevation of the ice sheets probably also influenced ice-sheet configurations elsewhere in the NH. For example, our hypothesised ice-sheet configurations for the last glacial cycle are consistent with the view that the development of substantial ice sheets in North America led to warming, and limited glaciation, in NE Asia during the LGM by altering atmospheric circulation patterns[33].

Spatial differences in the maximum extent of NH ice sheets between glacial cycles are also likely to have been caused by variations in moisture supply linked to complex ice-ocean-atmosphere interactions. For example, the larger extent of the EIS during MIS 6 compared to the maximum geographic ice-sheet extent during the last glacial cycle (MIS 2–5d) (Fig. 3b) has been attributed to wetter conditions over Eurasia during MIS 6, enabled by warmer global oceans[34]. Another, older example is the dominance of the Cordilleran Ice Sheet (CIS) compared to the smaller (and separated) Laurentide ice masses (Keewatin, Labrador and Baffin) during the late Gauss Chron (2.6–3.6 Ma; Fig. 1u), which has been attributed to the North American Cordillera blocking much of the north Pacific moisture from reaching the interior of North America during this time[35].

Notwithstanding the inherent uncertainties in producing these reconstructions, our hypothesised ice-sheet configurations clearly show the importance of topography in modulating the extent and rate of ice-sheet growth and decay. The EIS underwent the greatest magnitude of change in area between time-slices, increasing in area by over 1000% during the LGM relative to the warmer intervals of the last glacial cycle (MIS 3, 5a and 5c; Fig. 2a). Such huge expansion of the EIS during Mid- to Late Pleistocene cold periods reflects, in part, the much greater area of cold central Eurasia compared to warmer central North America.

The apparent susceptibility of the EIS to rapid and near-complete deglaciation (Fig. 2a) may be explained by the partially marine-based nature of this ice sheet, which covered the large epicontinental Barents-Kara Sea and North Sea during full-glacial periods[14,32,36]. Marine-based ice sheets, such as the present-day West Antarctic Ice Sheet, are more susceptible to rapid and potentially unstable ice-sheet collapse, for example through increased iceberg calving, in response to climatic and sea-level variations[37]. The Greenland Ice Sheet (GIS) and CIS have a comparatively small magnitude of variation in ice-sheet area between the reconstructed time-slices (Fig. 2a). Although some of our reconstructions are poorly constrained by empirical data, it is apparent that the relatively narrow continental shelf beyond Greenland and western Canada limits the maximum size that the GIS and CIS can attain.

**Sea-level equivalent ice volume.** Despite uncertainties, especially for older periods, our time-slice reconstructions clearly illustrate major fluctuations in ice-sheet extent (Fig. 1d–u) that generate a good fit with previously published global sea-level curves[38] (Fig. 2b). First-order estimates of the sea-level equivalent represented by the cumulative volume of our hypothesised ice-sheet reconstructions are produced using a simple area-volume scaling relationship (Methods). These cumulative ice volumes assume that the NH ice sheets reached their maximum extent at the same time and, therefore, are plotted at the times of lowest global sea level. As such, they should be viewed as the maximum amount of sea level lowering from NH glaciation. This assumption is compensated for, at least in part, by the fact that we do not account for the different densities of ice and sea water, which would produce an additional sea-level lowering of around 12%. We do not correct for the displacement of sea water by grounded ice because of uncertainties about long-term bathymetry and ice thickness. It should be noted that estimates of the eustatic sea-level equivalent are not fully independent for time-slices that were based, in part, on ice-sheet configurations from another time-slice (e.g. the EIS in MIS 16 and the LIS in MIS 8, 10, 12 and 16).

Again, and despite the large uncertainties, there is a particularly good fit between our best-estimate ice volumes and published sea-level records for glacial maxima, when geological evidence is often best-preserved (Fig. 2b). The sea-level equivalent volume of our LGM reconstruction (Fig. 2b), which is based mainly on an existing compilation of empirical evidence[14] (Supplementary Note 1), closely matches the c. 100 m sea-level equivalent for the NH ice sheets that has been estimated by other studies[39]. The discrepancy between this estimate and the c. 130 m of sea-level equivalent that is suggested by the benthic $\delta^{18}O$ stack (Fig. 2b) may be the result of potential inadequacies of current models in estimating glacial isostatic adjustments[39] as well as the exclusion of Southern Hemisphere ice masses from our study. There is also broad agreement for the four sub-stages of MIS 5 (a–d), although our best-estimates suggest that the NH ice sheets may have been slightly smaller than those of previous studies[38,40].

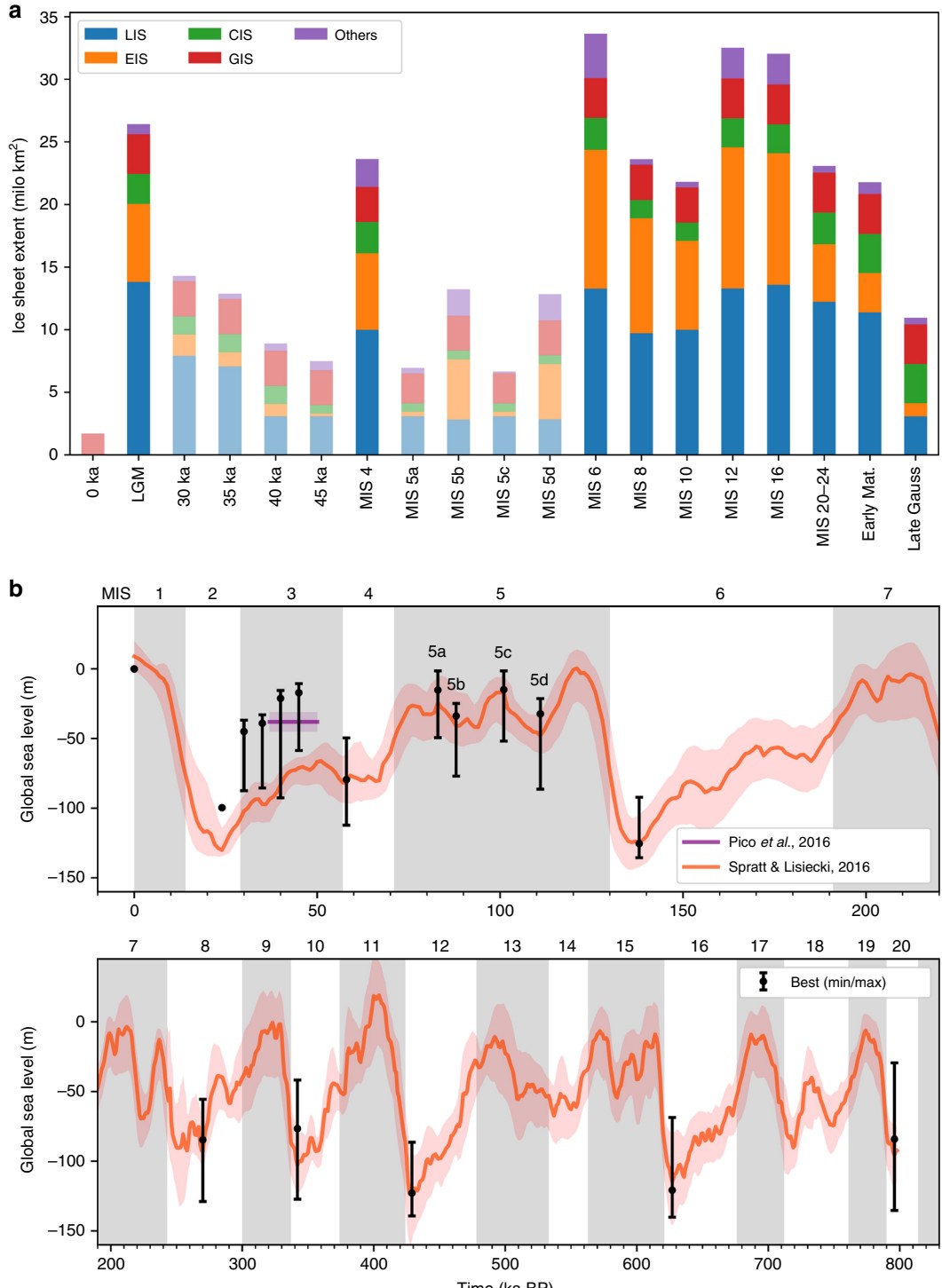

**Fig. 2** Extent and cumulative ice volume of NH ice sheets. **a** shows bar chart of ice-sheet extent at 18 time-slices through the Quaternary relative to present-day extent (0 ka), with each bar composed of individual ice-sheet extents. Bars with low-saturation colours are the comparatively warm intervals of MIS 3 and 5 and the present day, whereas high-saturation bars show the maximum ice-sheet extent during full-glacial periods. **b** shows the sea-level equivalent represented by the cumulative volume of the reconstructed NH ice sheets in this study (black bars), superimposed on previously published estimates of global sea level for the last 0.8 Ma. Black circles show the sea-level equivalent represented by our best-estimate reconstructions. Because our cumulative ice volumes assume that the NH ice sheets reached their maximum extent at the same time, our sea-level-equivalent estimates for the full-glacial periods of MIS 2, 4, 6, 8, 10, 12, 16 and 20–24 are plotted at the coldest point (lowest global sea level) within each of these time-slices. For the comparatively warm periods of MIS 5a and 5c, for which we attempted to capture the peak warmth, our sea-level estimates are plotted at the warmest point (highest global sea level) within these time-slices

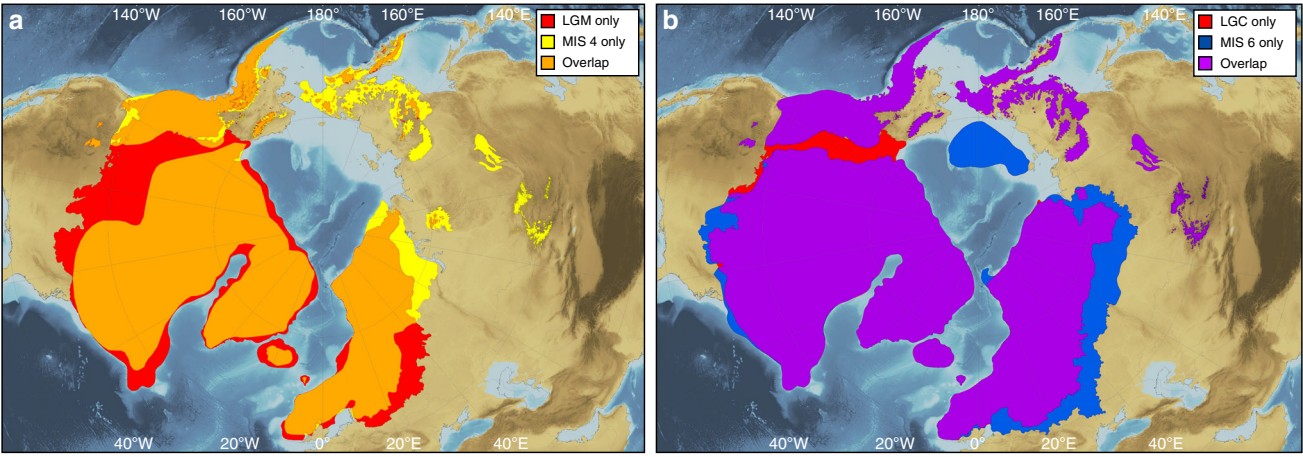

**Fig. 3** Comparison of NH ice-sheet extent during the last glacial cycle and MIS 6. **a** shows a comparison of the reconstructed ice-sheet extent during the LGM and MIS 4. The orange fill shows areas that were covered by ice sheets during both the LGM and MIS 4. **b** shows a comparison of the reconstructed geographical maximum ice-sheet extent during the last glacial cycle (MIS 2–5d) and MIS 6. The purple fill shows areas that were covered by ice sheets during the last glacial cycle (LGC) and MIS 6. Background is ETOPO1 1 arc-minute global relief model of the Earth's surface[72]

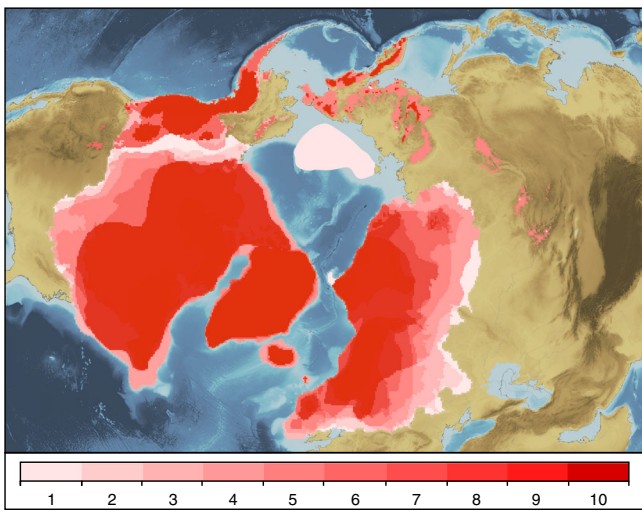

**Fig. 4** Intensity map of the number of times that each region was covered by ice sheets, produced by overlaying the best-estimate ice-sheet reconstructions from MIS 2, 3, 4, 5, 6, 8, 10, 12, 16 and 20–24. Regions shaded dark red were subject to glaciation 8–10 times through the last 1 Ma. Ice-sheet reconstructions for the early Matuyama (Early Pleistocene) and late Gauss magnetic chrons (Late Pliocene) are omitted because of the broad time-spans and high uncertainty of these reconstructions. Background is ETOPO1 1 arc-minute global relief model of the Earth's surface[72]

Our expectation is that future work might test and refine any discrepancies (e.g. at the local scale). Indeed, the one obvious discrepancy between our estimated ice volumes and the global sea-level curve occurs during MIS 3, when our reconstructions at four time-slices (45, 40, 35 and 30 ka) imply that ice sheets were considerably smaller and that, consequently, global sea level was substantially higher, possibly by as much as 30–40 m (Fig. 2b). To that end, we note that the sea-level curve derived by Pico et al.[40] is more consistent with our estimated MIS 3 ice-sheet volumes (Fig. 2b). Our reconstructions therefore support a growing body of evidence[11,41] that the NH ice sheets during MIS 3 may have been more limited in extent than previously thought, in the case of North America, or had almost entirely disappeared, in the case of Eurasia (Fig. 1e–h).

**Landscape evolution.** Combining our best-estimate reconstructions for the last *c.* 1 Ma (Fig. 4) shows the number of times that each region was covered by ice during the 10 time-slices since the late Early Pleistocene sampled in this study. To account for the different lengths of these time-slices, only the largest reconstruction within MIS 3 (which spans time-slices 30, 35, 40 and 45 ka) and within MIS 5 (which spans time-slices MIS 5a–d) were used. Areas that were ice-covered during the two oldest Late Pliocene to Early Pleistocene time-slices, the early Matuyama (1.78–2.6 Ma) and late Gauss (2.6–3.6 Ma) magnetic chrons, were not included because these span such long time periods. Although areas could have been ice-covered during additional glaciations, this map provides a useful conceptual framework to interpret the landscape evolution of the NH.

Regions shaded dark red were subject to glaciation 8–10 times through the last 1 Ma and were the main nucleation centres for the NH ice sheets[42] (Fig. 4). For most of these interior or core regions, ice-sheet development was probably linked to mountainous terrain (e.g. Alaska Range, Coast Mountains, east Baffin Island, Scandinavian mountains). For example, the LIS is known to have initiated over the Arctic/sub-Arctic plateaux of eastern Canada, where only small changes in temperature caused large shifts in the ratio between the accumulation and ablation areas of the ice masses[18]. The comparatively long history of ice-sheet occupation has had a pronounced effect on these landscapes that supported ice-sheet inception, which are generally characterised by terrain typical of enduring glacial erosion, including extensive areas of areal scour in low relief and selective linear erosion in high-relief coastal areas/fjords[43,44]. The erosion of regolith from these areas to expose harder crystalline bedrock with greater frictional resistance may have enabled Mid- to Late Pleistocene ice sheets to become thicker than their Early Pleistocene counterparts, contributing to the transition from predominantly low-amplitude, high-frequency (41 ka) ice-volume variations to high-amplitude, low-frequency (100 ka) variations under similar orbital forcings[45,46].

In contrast, regions shaded light red to pink represent areas covered by ice sheets during only the most extensive ice-sheet advances (Fig. 4). These, generally lowland, landscapes (e.g. Canadian Interior Plains, southern North Sea, southwest Russia, southern West Siberian Plain) typically exhibit ice-marginal features associated with glacial deposition and glaciofluvial reworking, including widespread and often thick glacial deposits and glaciotectonic features[47]. Although some of the older

ice-sheet reconstructions that informed Fig. 4 are based, in part, on ice-sheet extents from younger time-slices, there is empirical evidence for NH ice sheets reaching a southerly position between ~0.4 and 1 Ma (MIS 12, 16 and 20–24) that was similar to younger glaciations[24,26] (Supplementary Figures 8 and 9, Supplementary Tables 13–15). Locations where ice sheets reached the continental shelf-break during multiple Quaternary glaciations (e.g. Norwegian, Greenland, northern and eastern Canadian, and Barents-Kara Sea margins) are also key sites of glacial deposition, as indicated by major (up to 1 km-thick) glacial-sedimentary depocentres, or trough-mouth fans, on the continental slope[48,49]. Ice advance also had a profound impact on continental hydrology and drainage patterns through the Quaternary. For example, in both North America and Eurasia, the formation of large ice-dammed lakes led to the re-routing of major drainage systems[36,50], which affected climate and ocean circulation[51]. We hypothesise recurrent advances of the LIS and EIS to a similar position during several glaciations prior to the LGM (e.g. MIS 5d, 6, 12, 16) (Fig. 4), implying that proglacial lakes filled and drained repeatedly during earlier glacial periods.

## Discussion

This paper and the accompanying online database provide a synthesis of empirical data and modelled outputs relating to pre-LGM ice sheets, and should be viewed as new hypotheses relating to the likely ice-sheet extent at key time intervals through the Quaternary (Fig. 1, Supplementary Figures 2–10). Our maps clearly highlight the varying spatial and temporal distribution of empirical evidence for pre-LGM ice sheets, and provide hypotheses of best-estimate ice-sheet configurations to be tested by future empirical and modelling efforts. The spatial differences in ice-sheet configuration that are illustrated both within and between glacial cycles (Fig. 3) illustrate the importance of using pre-LGM ice-sheet extents as input to earth systems and global climate models that span the Quaternary, and demonstrate the need to fully understand and model the time-transgressive nature of ice-sheet margins within glacial cycles. Further work incorporating ice-volume and ice-loading histories could usefully examine glacio-isostatic effects on relative sea level or how ice-sheet thicknesses perturb atmospheric circulation patterns. Our ice-sheet outlines could also be used to reconstruct the evolution of major proglacial lakes and changes in the routing of surface runoff through time[52].

The reconstructions also provide a dataset for further analysis of the development of mid- to high-latitude permafrost and vegetation changes during the Quaternary. In particular, the extent of glacial ice fringing Beringia potentially played a key role in determining the level of faunal and floral interchange between Eurasia and the Americas. Whilst this pathway was only closed during the LGM (because of the coalescence of the LIS and CIS: Fig. 3), the ice-sheet margins at other times, and the consequent climatic conditions in Beringia itself and in the corridor to North America, affected the role of this region as a refugium as well as the level of exchange with the American interior[53]. With the increased ability to reconstruct changes in geneflow among populations using genomic data[54] the diachronic view of the pan-Beringian connection provided by our reconstructions offers a context to understand the ebb and flow of movement between Eurasia and the Americas by different species.

## Methods

**Data compilation**. The data (empirical evidence and numerical model outputs) were compiled through a literature search of published evidence for the spatial extent of NH Quaternary glaciation. Details of the source publication, methodology and age of glaciation were entered into a database (Supplementary Tables 1–17). Except for noting the error bounds for the reported age of glaciation derived from each publication, we do not assess the validity for each data source. We do this to be as transparent as possible in our methods and to avoid a further level of subjectivity.

Our database includes evidence for NH glaciation that falls into 17 time-slices. These are: 30, 35, 40 and 45 ka, MIS 4 (58–72 ka), 5a (72–86 ka), 5b (86–92 ka), 5c (92–108 ka), 5d (108–117 ka), 6 (132–190 ka), 8 (243–279 ka), 10 (337–365 ka), 12 (429–477 ka), 16 (622–677 ka) and 20–24 (790–928 ka), the early Matuyama palaeo-magnetic Chron (1.78–2.6 Ma), and the late Gauss palaeo-magnetic Chron (2.6–3.59 Ma) (Fig. 1). The bounding ages for each time-slice are from Railsback et al.[22] These time-slices were chosen to reflect the varying amount and resolution of the available evidence for glaciation extent through the Quaternary.

Our literature search was based on the following general principles. We mapped the changing ice-marginal position of the ice sheets and therefore did not include data points that are located well inside a suggested ice margin. In cases where the same author(s) have published multiple reconstructions for the same area, we used the most recent hypothesised ice-sheet extent. When using ice-sheet outlines that are derived from a synthesis of previously published empirical evidence[32,55], we did not include all of the data points that informed the synthesised reconstruction. It is beyond the scope of this study to review all marine-sedimentological evidence for ice sheets (i.e. ice-rafted debris (IRD)). Sedimentological and stratigraphic data (including marine seismic data) were used to supplement empirically derived and numerically modelled ice-sheet outlines and were particularly targeted to the oldest time-slices (early Matuyama and late Gauss palaeo-magnetic Chrons), for which published ice-sheet outlines are scarce.

We did not compile data for the ice extent at the relatively well-defined LGM, around 26.5 ka[13]. Rather, a best-estimate reconstruction was derived mainly from the compilation of Ehlers et al.[14], with modification of the ice-sheet limits in some areas (Fig. 1d, Supplementary Note 1). With the exception of MIS 3, 5a and 5c, for which some empirical data are available[56–58], we do not provide ice-sheet reconstructions for interglacials/interstadials because of a paucity or absence of reported evidence for glaciation during these periods.

Some empirical outlines and data points are included in more than one time-slice; for example, where the error bounds of an age estimate span multiple time-slices or where an age estimate lies on the boundary between two time-slices. For modelling results in which many reconstructions are available for each time-slice, we used the least extensive reconstruction (i.e. peak climatic warmth) for the relative warm intervals (e.g. MIS 5a and 5c), and the largest reconstruction (i.e. peak climatic coldness) for all other time-slices.

**Outline digitisation**. Data on the extent of glaciation during the Quaternary were digitised and georeferenced using Esri's ArcGIS software. Three types of data were digitised: empirical outlines of ice-sheet extent (coloured fill), which are often regional or ice-sheet wide; modelled outlines of ice-sheet extent (coloured lines), which are typically ice-sheet wide or span the NH; and point-source data (red circles) that show the former occupation of a site by ice (Supplementary Figures 2–10). We include published evidence for mountain glaciers, ice fields, ice caps and ice sheets in the raw data maps. Some empirical outlines were too small to georeference and were plotted as point-source data. Some of our raw data maps show more than one data point (red circle) for each previously published study; for example, where there are multiple data sites. Where many dates have been acquired from a relatively small area, we show a single data point in a representative location. Ice-marginal positions that are inferred from studies of IRD in sediment cores were included as point-source data. In these cases, the data point (red circle) was placed at the position that the core was taken, and an arrow shows the location that the ice was interpreted to have reached. Unless a glacial curve diagram is also included, the presence of IRD in a marine sediment core is taken to indicate that the ice sheet reached close to the present-day coastline. Only grounded ice sheets were mapped; we do not depict ice shelves, e.g. in the Arctic Ocean[59]. We did not plot the locations of areas in which the absence of glaciation has been inferred. However, information on ice-free areas informed the best-estimate reconstructions and is included in the explanations that accompany the maps (Supplementary Notes 1–18).

Our raw data maps (Fig. 1a, Supplementary Figures 2–10) were designed to be as objective as possible. No smoothing function was applied to the digitised outlines. As such, inaccuracies may have been inherited from the original data source and/or may originate from the digitising and georeferencing process. The raw data maps show the amount and distribution of published evidence for the general extent of the NH ice sheets during each time-slice: they should not be used for local-scale studies or as a substitute for the original source data.

**Maximum, minimum and best-estimate reconstructions**. We used a consistent methodological framework to produce maximum, minimum and best-estimate hypotheses of ice-sheet extent from the maps of previously published ice-sheet extents (Supplementary Figures 2–10). This approach builds upon that of Hughes et al.[11], whose reconstructions of ice-sheet extent used maximum and minimum limits to represent uncertainty. The use of maximum, minimum and best-estimate reconstructions in our study provides a visual indicator of uncertainty and identifies regions and time-slices where future work should be directed.

Although mountain glaciers, ice fields and ice caps developed in many high-relief areas of the NH during the Quaternary, including the Himalaya, the

European Alps and the Rocky Mountains[60–62], our maximum, minimum and best-estimate reconstructions were only performed for areas that have been suggested to have been covered by ice masses > 50,000 km² (i.e. ice sheets). This is because of the broad, hemispheric scale, focus of our reconstructions and their implications for global sea level, as well as the uncertainties involved in reconstructing the extent of mountain glaciation through the Quaternary. The present-day ice cover is incorporated into our reconstructions in all cases apart from the minimum reconstructions for the relatively warm periods of 45 ka, MIS 5a, MIS 5c and the late Gauss palaeo-magnetic Chron.

Our reconstructions aim to capture the maximum extent of each ice sheet within each time-slice, with the exception of the comparatively warm periods of 45 ka, MIS 5a and 5c for which we attempt to capture the peak warmth. The maximum extent of glaciation may have occurred at any time(s) within a time-slice; for example, for the long late Gauss palaeo-magnetic Chron (2.6–3.6 Ma), the maximum extent of the EIS probably occurred close to the youngest part of the time-slice, around 2.6–2.7 Ma. We do not capture variations in ice-sheet extent within a time-slice. For example, in the early Matuyama palaeo-magnetic Chron (1.78–2.6 Ma), our best-estimate reconstruction does not show evidence for a reduced GIS during an Early Pleistocene warm period around 2.4 Ma[63,64].

Details about the decisions made in reconstructing the maximum, minimum and best-estimate ice-sheet extents for each separate time-slice are provided in Supplementary Notes 1–18. In general, we used the empirical data where they are available, and the modelled ice-sheet extent where empirical data are lacking. Detailed outlines were generally followed over coarser outlines, and we took the smaller ice-sheet option when uncertain.

For regions and/or time-slices where empirical and modelled data are not available, a feasible ice-sheet extent was derived using the ice-sheet configuration from another time-slice that has a similar value in the global δ18O record[1] (Supplementary Notes 1–18). It should be noted that, in these cases, estimates of the eustatic sea-level equivalent represented by the cumulative volume of the ice sheets are not fully independent. This mainly affects the best-estimate reconstructions for the EIS in MIS 16 and the LIS in MIS 8, 10, 12 and 16. In some time-slices, we used the best-estimate reconstruction from another time-slice to constrain the maximum ice-sheet extent; for example, for the maximum reconstruction of the EIS at 40 ka, we followed the maximum modelled ice-sheet extent but did not allow this to be larger than the best-estimate LGM. Given the necessary uncertainties that arose from this exercise, robustness scores were developed to rank the reliability of each reconstruction (see below).

To avoid unnecessary complexity, several ice-sheet templates were used for the ice extent in the North American Cordillera, Greenland, Iceland and NE Asia. There are six configurations for the CIS. The first configuration is the maximum Quaternary (pre-Reid) extent in Alaska[65] and the Yukon[66], combined with modelled MIS 6 outlines[17,19] for the southern CIS margin. This outline is extended to the south to include ice in the Cascades and Rocky Mountains of North America, as in the LGM ice-extent template. The second configuration is the Reid limit of suggested MIS 4/MIS 6 age[65,66]. This outline is extended to the south to include ice in the Cascades and Rocky Mountains of North America, as in the LGM ice-extent template. The third configuration is the LGM ice-sheet extent of Ehlers et al.[14], which is simplified in the central North American Cordillera. The fourth configuration is the regionally modelled ice-sheet extent at 30 ka from Seguinot et al.[67] This outline is reduced slightly at its southern and eastern margin so that it does not extend beyond the LGM ice-extent template. The fifth configuration is schematic coastal mountain glaciation. The sixth configuration is undefined mountain glaciers (no outline). In order to calculate the area of each ice sheet, the maximum reconstruction for the CIS was used to define the boundary between the CIS and the LIS.

In Greenland, there are four ice-sheet configurations. The first configuration shows the ice sheet at the shelf-break. The second configuration shows the ice sheet on the inner- to mid-shelf. Because of the narrow continental shelf around parts of Greenland, we do not differentiate between an inner-shelf or mid-shelf position. The third configuration is the present-day coastline. The fourth configuration is the present-day ice extent. There are three ice-mass configurations for Iceland. The first configuration shows shelf-break glaciation. The second configuration shows the ice sheet at the present-day coastline. The third configuration is the present-day ice extent. There are three configurations for ice masses in NE Asia. The first configuration is a combination of two reconstructions of maximum Quaternary ice-sheet extent[30,31]. The second configuration is the ice-sheet extent at the LGM[31]. The third configuration is undefined mountain glaciers/no ice sheet (no outline).

It is interesting to note the generally poor alignment of the published numerical modelling results with the empirical evidence (Supplementary Figures 2–10). There are no clear patterns in terms of regions in which the models performed better or worse, and the models often show ice-sheet extents that are unfeasible (i.e. are beyond the all-time Quaternary maximum). Modelled ice-sheet limits are therefore not incorporated in most of our reconstructions. This further demonstrates the need for information about the extent of the Quaternary ice sheets to be used as input to earth systems and global climate models. Some of the models included in our compilation have been constructed or calibrated using existing empirical data about the ice margins and/or benthic δ18O stack, as described in Supplementary Tables 1–17. Although such model outputs could produce some circular reasoning, we note that our best-estimate hypotheses are rarely informed only by modelled outlines.

Our ice-sheet reconstructions do not capture the time-transgressive nature of the ice-sheet limit between different regions of the NH prior to the last glacial cycle (MIS 2–5d). Different ice masses reached their respective maxima at different times during the last glacial cycle; for example, ice in northern Eurasia and mountain glaciers in mid- to high-latitudes reached their maximum early in the last glacial cycle, whereas most of the LIS and the southern EIS reached their maximum close to the global LGM[9]. This pattern is likely to have also existed for ice sheets in older Quaternary glacial periods[15]. The mapping of time-transgressive ice margins, however, can only be achieved through the development of techniques to date older sediments at sub-stage resolution.

Our maps of ice-sheet extent through the Quaternary show the sea level and topography of the present day (Fig. 1d–u). This is because of the uncertainty involved in calculating isostatic adjustments and rates of sediment erosion during the Quaternary. We recognise, though, that NH topography has changed significantly during this time, including as a consequence of the glacial erosion of mountain ranges and the progradation of the continental shelf through sediment delivery to marine margins[68].

Overall, our maximum, minimum and best-estimate reconstructions are necessarily subjective, but they provide the first systematic and consistent approximations of generalised NH ice-sheet extents through the Quaternary.

**Robustness scores**. To aid interpretation of our maps, each best-estimate ice-sheet reconstruction has been allocated an overall robustness score (from 0 to 5) (Fig. 1d–u, Supplementary Figures 2–10). This score represents an average of the individual scores for each of the four main ice-sheet regions (EIS, LIS, CIS and NE Asia) during that time-slice. The robustness score for each ice sheet is a subjective assessment of the amount and reliability of the source data from which the ice-sheet extent was constructed. The scores are broadly defined as follows. First, a robustness score of 0 shows that no empirical or modelled data from this region are available for this time-slice; the ice-sheet extent is taken from a time-slice with a similar value in the global δ18O record[1]. Secondly, a robustness score of 1 suggests that modelled data are available and the ice-sheet extent may have been produced, in whole or in part, from a time-slice with a similar value in the global δ18O record[1], or the ice-sheet extent at another time-slice may be used to constrain a modelled outline. Thirdly, a robustness score of 2 indicates that point-source empirical data or localised empirical outlines are available to inform the ice-sheet extent. The ice-sheet extent at another time-slice may inform some of the reconstruction. Fourthly, a robustness score of 3 suggests that local empirical outlines or regional empirical outlines of contrasting extent inform the ice-sheet reconstruction. Fifthly, a robustness score of 4 suggests that a significant portion of the reconstructed ice-sheet margin is derived from empirical outlines. Finally, a robustness score of 5 suggests that almost all of the reconstructed ice-sheet margin is derived from empirical outlines that are in broad agreement.

The robustness scores of individual ice-sheet reconstructions vary considerably between time-slices. Lower scores are generally allocated to older time-slices, interstadial periods (e.g. 45 ka, MIS 5a and 5c), and glacial periods such as MIS 8 and 10 that occurred between glaciations of larger extent. These are the time-slices in which empirical evidence is typically poorly preserved and when modelling efforts are generally lacking.

**Area-volume scaling**. We utilised a scaling power law that converts area (A) to volume (V) to estimate the contribution of individual NH ice sheets to global sea-level changes (i.e., eustatic sea-level changes) (Fig. 2b). The equation for the area-volume scaling is

$$V = cA^\gamma. \tag{1}$$

For the scaling exponent $\gamma$, 5/4 ( = 1.25) is a widely accepted value for ice sheets[69]. The coefficient $c$ was derived from outputs of three numerical ice-sheet modelling studies[17,19,70], which have been used previously to synthesie pre-LGM NH ice-sheet configurations. For each ice sheet, different coefficients were calculated and are shown in Table 1. The agreement amongst the numerical models is best for the

**Table 1 Area-volume scaling coefficients, c, for the different NH ice sheets as calculated from the output of three numerical ice-sheet modelling studies**

|        | de Boer et al., (2014)[19] | Ganopolski and Calov, (2011)[17] | Zweck and Huybrechts, (2005)[70] | Mean | Std |
|--------|------|------|------|------|------|
| CIS    | 0.78 | 0.78 | 1.40 | **0.99** | **0.36** |
| EIS    | 0.89 | 0.88 | 0.77 | **0.85** | **0.07** |
| LIS    | 0.92 | 0.98 | 0.94 | **0.94** | **0.03** |
| GIS    | 0.92 | 1.12 | 1.17 | **1.07** | **0.14** |
| Others | 0.38 | 0.58 | 0.71 | **0.56** | **0.17** |

EIS and LIS, i.e., the standard deviation is smaller than 10%. Coefficient uncertainties are larger for the CIS, GIS and ice masses in NE Asia. However, compared to the EIS or the LIS, the overall area of these ice sheets is relatively small (see Fig. 2a) and so is the corresponding ice-sheet volume.

Using equation (1) with the ice-sheet-specific scaling coefficient, $c$, the ice-sheet volumes and corresponding global sea-level contributions were calculated for each of the synthesised pre-LGM time-slices. For each time-slice, the volume for the best, minimum and maximum area estimates were translated into global sea-level change by dividing the ice-sheet volume by the area of the world's ocean, i.e., ~362 million $km^2$. The final values for the best, minimum and maximum global sea-level contribution of the NH ice sheets are shown in Fig. 2b. Note that for each stage prior to 45 ka, with the exception of MIS 5a and 5c for which we attempt to capture the peak warmth, the global sea-level contribution is placed at the global sea-level lowstand in the stage. This decision was made based on the fact that the ice-sheet areas for each stage correspond to the aggregated maximum areas within that stage and not to the instantaneous ice-sheet extent at a specific point in time.

Maximum global sea-level contributions of Antarctic ice sheets are not included in this study. The volume of these ice sheets prior to the LGM remains subject to large uncertainties and published estimates range between 10 and 35 m, depending on the method applied[21,71]. However, if we assume that Antarctica's sea-level contribution is linear with global sea-level changes, this value would have the same order of magnitude as the uncertainty that is associated with the Lisiecki and Raymo[1] dataset, which is between 5 and 22 m.

The estimates of eustatic sea-level for MIS 8, 10 and 20–24 are not fully independent because parts of the ice-sheet extent for these periods were derived from the ice-sheet configuration during MIS 4 (which has a similar value in the global $\delta^{18}O$ record to MIS 8, 10 and 20–24[1]). However, we note that our estimates of eustatic sea-level for MIS 3, which are up to 30–40 m higher than most previously published sea-level curves (Fig. 2b), are derived independently from the sea-level record.

## Data availability
All maps and data sources are shown in Supplementary Figures 2–10 and Supplementary Tables 1–17. Shapefiles of our reconstructions, as well as the digitised and georeferenced empirical and modelled data, are available on the Open Science Framework [https://osf.io/7jen3/].

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

## Acknowledgements

During this work, C.L.B. was in receipt of a Junior Research Fellowship at Newnham College, University of Cambridge, and a grant from the Norwegian VISTA programme. MM was supported by a Swedish Research Council International Postdoctoral Fellowship (No. 637-2014-483). This work was supported by an ERC Consolidator Grant to AM (Local Adaptation 647787). The Leverhulme Foundation is also thanked for financial support to PLG, and we acknowledge funding from the DIFeREns2 Junior Research Fellowship (No. 609412; Durham University) to ASD. We thank René Barendregt for providing information about North American ice sheets. We also thank Richard Gyllencreutz for his helpful review of this manuscript.

## Author contributions

A.M. devised the project together with J.B.M.; C.L.B., D.K.M. and M.M. reviewed the literature with input from M.K., A.S.D., P.L.G., C.R.S. and J.B.M.; C.L.B. drew the outlines with help from M.M. and input from the other authors; M.K. devised and implemented the conversion from area to volume; C.R.S., J.B.M. and A.M. wrote a first draft of the paper which was improved by input from all other authors; C.L.B. wrote the supplementary information and methods, with input from all other authors.

## Additional information

**Competing interests:** The authors declare no competing interests.

