## [Peer Review File · Nature Communications]

Reviewers' comments:

Reviewer #1 (Remarks to the Author):

The authors present a temporal succession of the Northern Hemisphere ice-sheets coverage that consists of 17 discrete time snapshots spanning the last ~1.0 Ma. Each time slice stems from the synthesis, digitization and georeferentiation of published empirical evidences (data points and outlines) and numerical models. The authors adopt a methodological framework where decisions are made according to a ranking scheme that generally gives priority to data over models. Each reconstructed temporal snapshot is accompanied by a minimum, maximum and best estimate and is ranked from low to high on the basis of a robustness score. The latter depends on the availability and agreement between empirical data and models and is characterized, to some degree, by unavoidable subjectiveness.

The results show that NH ice sheets were characterized by significant spatial differences and asymmetry throughout the Quaternary. The reconstructed global ice-sheets volumes are generally consistent with the eustatic curve based on delta-18O curve but are significantly smaller during MIS 3, as supported also by sedimentological observations (Pico et al., 2016). The authors point out that there is a general mismatch between the ice sheets margins that are based on empirical evidences and the outcome of numerical models. The proposed methodology and its outcome could therefore add additional and necessary constraints to the numerical models.

This paper does not present novel data or model output, but a longed-for capitalization on existing empirical data and numerical models that, through a sound and convincing methodological framework, provides input for a large scientific community. Several disciplines such as biogeography, paleoclimatology, paleoceanography and paleontology will benefit from this research.

In particular, ice-sheets modellers will be stimulated to combine the reconstructed NH ice-sheets margins with paleo RSL indicators and use them as constraints in their algorithms. The most advanced ice-sheets models do currently account for glacial and hydro isostasy (GIA) and will soon include sediments isostasy. Therefore, the inclusion of realistic ice-sheets margins as constraints would definitely contribute to a full transition from "ice-sheet models" towards actual "Earth systems models".

Also, realistic NH ice-sheets margins are a necessary when the GIA-driven RSL changes across interglacial are to be evaluated and removed in order to constrain the Greenland and Antarctic Ice Sheets retreats (see all the recently published works on MIS 5e).

Overall, the manuscript is very well written, it accounts for a comprehensive review of existing data and models and is accompanied by clear and exhaustive Methods and Supplementary Notes. The latter allow for a straightforward reproduction of the presented results. The figures and the Extended data are clear.

Comment/request for the authors:

Several models are considered in this work. Among these, some have been constructed by including available ice-sheets margins based on empirical data (see ICE-5G from Peltier (2009) and ANU from Lambeck et al. (2006)). Do the authors think that there could be a bias due to this sort of circular problem? The same holds for the ice sheet models that are forced by the delta-18O curve (see de Boer et al., 2014). Can the authors mention in the Supp. Notes whether a model is based on existing empirical outlines?

Typos and errors:

Main text

1. Missing year in reference 36, Spratt et al. (line 355)
2. Misspelled name of Lisiecki in Figure 2b, legend (change Liesicki)

Supp. Notes, References:

1. Missing Year in Eccleshall et al., (line 1268)
2. Missing Year in Gozhik et al., (line 1293)
3. Missing Title in Kleman et al., (line 1363)
4. Missing Year in Montelli et al., (line 1428)
5. Missing Title in Nikolskiy et al., (line 1439)
6. Missing Year in Rea et al., (line 1469)

Reviewer #2 (Remarks to the Author):

Review NCOMMS-18-38606

Title: The configuration of Northern Hemisphere ice sheets through the Quaternary

General comments

This manuscript presents ice sheet reconstructions for major paleo-ice sheets in the Northern hemisphere through the Quaternary. The times decided for reconstruction are pragmatically chosen based on research interest and data availability, and represent every 5 ka from LGM to 45 ka + MIS 5abcd, 6, 8, 10, 12, 16, 20-24, and for the paleomagnetic chrons Early Matuyama and Gauss. The timings for the MIS-time slices are based on Railsback et al. (2015), which is very sensible to avoid misunderstandings of MIS notation and/or timing. The 17 reconstructions are based on a comprehensive synthesis of published data from empirical ice sheet-wide reconstructions, empirical point data, and where empirical data is lacking, from numerical model results. The reconstructions are supplied with maximum, best-estimate, and minimum extents and a robustness score, which although somewhat subjective greatly facilitates assessment and use of them.

The manuscript is very well written, concise and has a clear structure. The main manuscript is supported by a separate Methods chapter, and Extended Data with larger and more detailed figures to be used together with Supplementary Notes containing all relevant metadata about the data used and the rationale behind the interpretations. The manuscript contains 4 figures, which are all relevant and well designed to be as small as possible while clearly showing what they are aimed for. Some details about the figures are given below.

The main conclusions from their analysis of the new reconstructions are:

- There is large ice sheet asymmetry between different glacials
- The Eurasian Ice Sheet showed the largest asymmetry
- The Laurentide Ice Sheet was much reduced during MIS 3, implying 30-40 m higher sea levels at that time than previous estimates, which is supported by recent MIS 3 sea level data from Pico et al. (2016).

The conclusions are reasonable and they are well supported by the data presented. However, it is the reconstructions (maps in Fig. 1 and Extended Data) that constitute the main contribution of this manuscript.

This manuscript presents, through a consistent methodology, the most comprehensive and accurate synthesis of NH Quaternary Glaciation extent to date, and will be of great value to many fields of research, including paleoclimatology, paleoglaciology, numerical modelling of ice sheets and climate, archaeology, and paleontology.

Specific comments

Main Manuscript

(line numbers refer to PDF version. Word-document was numbered differently)

Line 157: "... good fit with previously published global sea-level curves ...". It may be relevant to mention that the presented LGM reconstruction matches the result of ~100 m sea level equivalent for the northern Hemisphere by Simms et al., 2019 (QSR, <https://doi.org/10.1016/j.quascirev.2018.12.018>). Simms et al. study the mis-match between ice sheet and paleo-sea-level reconstructions, and conclude that ~15 m of sea level appears "missing" in the LGM reconstructions.

Line 162 (see also Fig 2b below): "These cumulative ice volumes do not include Southern Hemisphere ice masses and assume that the NH ice sheets reached their maximum extent around the same time and, therefore, should be viewed as a maximum sea-level lowering." This is confusing for two reasons: 1) the wording and 2) the assumption. 1) Because the amount of lowering is minimum when the sea level is at maximum (i.e. at its shallowest). 2) If the NH ice sheets DID NOT reach their maximum at the same time, the amount of sea level lowering would be smaller (i.e. the sea level would be higher), making the reconstruction a MAXIMUM amount of lowering. However, if the SH ice sheets WERE included, the amount of sea level lowering would be larger (i.e. the sea level would be lower), making the reconstruction a MINIMUM amount of lowering. Thus, the reconstructed sea level is neither a minimum nor maximum, just approximate. As there is no way around this problem but to include SH ice sheets and/or make synchronous time slice maps, I suggest that the end of this sentence is rewritten as something like "... and, therefore, the sea levels are plotted at the times of lowest global sea level, and represent the maximum amount of sea level lowering from NH glaciation".

Figure 2b (see also Line 162 above): "Because our cumulative ice volumes assume that the NH ice sheets reached their maximum extent around the same time, our sea-level-equivalent estimates for the full-glacial periods of MIS 2, 4, 6, 8, 10, 12, 16 and 20-24 are plotted at the coldest point (lowest global sea level) within each of these time-slices.". First, "around the same time" should be "at the same time". Second, here they use the same statement as on Line 162 (simultaneous NH ice sheet maximum in the reconstruction), but use it to explain where on the timescale the calculated sea levels are plotted. This is reasonable, and is also stated on Line 170-173 in Methods.

Main manuscript Figures

Fig. 1a: the symbol for “Outline from empirical data” could be made multi-colored (e.g. divided in 3 fields with orange, light blue and light green) to be clearer.

Fig. 2: All labels should be slightly bigger, so the figure could be printed smaller. 2a) The x-axis labels would be easier to connect to the correct bar if they were moved slightly to the left (so they point towards their respective bar instead of being centered under it). 2b: Please add minor tick marks for every 10 ka on the lower x-axis.

Fig. 3: All labels should be slightly bigger, so the figure could be printed smaller. Caption: The ETOPO1 dataset should be cited as “Amante, C. and B.W. Eakins, 2009. ETOPO1 1 Arc-Minute Global Relief Model: Procedures, Data Sources and Analysis. NOAA Technical Memorandum NESDIS NGDC-24. National Geophysical Data Center, NOAA. doi:10.7289/V5C8276M”. Please put this in the reference list, and refer to it as Amante et al. (2009) in the caption.

Fig. 4: The intensity map is a great idea. However, I don’t see the point of translating the number of glaciations per area to a “glaciation intensity value” of 0 to 1 with a smooth color scale, as the glaciation intensity on the map is not continuous. It would be clearer if the legend showed discrete boxes of the corresponding colors from the map associated with the number of glaciations covering each area from 0 to 8 (or >8? The text says >8 but that’s not visible in the map). This map has large enough labels for the figure to be made smaller, but then the legend needs to be larger (could be placed over southern Eurasia instead).

Methods

The referencing of the methods chapter is based on numbers in the reference list (not alphabetical), but references are not numbered in the text. Please make this coherent.

The robustness scores are clearly defines and well justified, and are strengthening the manuscript.

For the ice sheet outline digitization, several assumptions and base postulates are given. All are reasonable and necessary. I comment on the most, in my view, important ones below:

- locations of ice free-indicating points are not plotted, but were used to inform the reconstructions and described in the Supplementary Notes. (Line 41-44). This is a necessary descision, because most ice-free indications are redundant (only ice-free points close to other points overridden by ice are relevant), and plots would be cluttered if all were included.

- “...the raw data maps [...] should not be used for local-scale studies or as a substitute for the original source data.” (Line 49-50). This ensures that original publications get credit and discourages over-interpretation of the data on a local scale; this work regards ice-sheet wide extent.

- only ice masses > 50 000 km² were considered (Line 60-62). This is a reasonable cut-off for a hemispheric-scope reconstruction, i.e. excluding ice caps smaller than about the size of Denmark.

- " For regions and/or time-slices where empirical and modelled data are not available, a feasible ice-sheet extent was derived using the ice-sheet configuration from another time-slice that has a similar value in the global $\delta^{18}O$ record (Spratt and Lisiecki, 2016) (Supplementary Notes). It should be noted that, in these cases, estimates of the eustatic sea-level equivalent represented by the cumulative volume of the ice sheets are not fully independent." (Line 68-73). This is a clever and relatively transparent way of filling data gaps, although it makes calculations of sea-level contributions from those ice caps circular and forces the sea level contributions in the direction to fit the Spratt and Lisiecki (2016) curve better. The authors are aware of this.

Line 53-55: Please include a reference to Hughes et al. (2016) to acknowledge that their ice sheet reconstruction was the first to provide maximum, minimum and best-estimate lines for ice sheet extent. This was an important conceptual improvement in the usefulness (especially for modellers and for identification of areas/times where data is lacking) and credibility assessment of ice sheet reconstructions, which should be acknowledged.

Line 167-168: " For each time-slice, the volume for the best, minimum, maximum area estimates were translated into global sea-level change by dividing the ice-sheet volume by the area of the world's ocean, i.e., ~362 million km²." The authors are thus not compensating for the different ice and seawater densities of 917 and 1028 kg m³ in calculating their sea level equivalents (SLE). The ice-to-water density ratio is 0.892, so neglecting this gives only 89.2% of the amount of sea level lowering (or equivalently, the sea level should be lowered by another 12.1%). Neither do they state whether they assume that sea water replaces ice where ice was grounded on the sea floor, which is important because buoyancy will cause marine ice sheet parts to not contribute with their full volume to the sea level change. This effect is minor for shallow water depths (e.g. about 1 m SLE for the Barents-Kara ice sheet at LGM at 210 m mean water depth; Hughes et al., (2016)), but may be important for the proposed grounded ice in the deep Arctic Ocean for MIS 4, 5b, 5d, 6, 12, 16, 20-24, Early Matuyama, and Late Gauss.

Supplementary Notes

The supplementary notes are to be read together with the figures in Extended Data.

There are several occasions of statements like "Due to uncertainty [a certain extent] is used for [a certain part of an ice sheet]", on Line 133, 148, 188, 238, 358, 421, 554, 610, 689, 694, 700, 703, 753, 763, 759, 933, 940, 1006, and 1077. What is uncertain is only specified in 3 of these (610, 763 and 940). Please specify wherein the uncertainty lies for all these occasions, as they are the rationale for

using some assumed ice extent in these times/locations. In most cases this is probably because no empirical data exist for this location at this time, but it should be explicitly stated if that is the case.

MIS 6 Table: Graphical comment - the blue line in the legend for "Colleoni et al., 2016" is misplaced and should be moved slightly to the left.

Line 721-723: "... but the three ice advances that are proposed between 0.2 and 0.75 Ma most likely occurred in MIS 6, 12 and 16." Why most likely at those times? I suppose this is based on the fact that MIS 6, 12, and 16 are the most extensive glaciations in this time span according to the global sea level record of Spratt & Lisiecki (2016), but the reason should be written.

Early Matuyama Table and Explanatory text Line 1014, and Late Gauss Table: Data for the LIS is used with reference to "Barendregt pers. comm.". As shown in Extended Data Figure 8c and 8d, the entire North American ice sheet complex, except for a small patch at the southernmost extension of the LIS in Early Matuyama, is based on "Barendregt pers. comm.". It would not be acceptable to base an interpretation of a continent-wide glaciation on unpublished and undisclosed data – how does the reader know the data even exists? However, the Barendregt data is actually not used (or at least seems redundant) – the reconstruction for Early Matuyama is (as far as one can discern in the printed figures) based entirely on the maximum MIS 6 outline and Balco & Rovey (2010), and for Late Gauss is based on Kleman et al. (2010) and Kaufman et al. (2011). Either make a case that the Barendregt data is important, and preferably include it as a table, or omit it from the manuscript. You can mention in the supplementary text that your interpretation is supported by the unpublished Barendregt data, but not that it is based on it.

Supplementary Notes References

Line 318: Mangerud (2011) – is this Mangerud et al. (2011)?

Line 1363: Kleman et al., 2008 – the title of the paper "Patterns of Quaternary ice sheet erosion and deposition in Fennoscandia and a theoretical framework for explanation" is missing.

Richard Gyllencreutz

Department of geological sciences, Stockholm university

Reviewer #3 (Remarks to the Author):

The manuscript “configuration of Northern Hemisphere ice sheets through the Quaternary” presents 17 pre-LGM ice sheet reconstructions based on a compilation of existing data and models. In addition, they calculate ice volume and discuss various mechanisms to explain the apparent ice sheet asymmetry in build-up and decay of sheets during the Quaternary. It has clearly been a large effort to compile the data. The research idea is excellent and the potential outcome of this exercise could potentially reveal something ground-breaking about the NH ice sheet evolution in the Quaternary that would be suitable for Nature Communications.

However, in its present form I cannot recommend publication as there are a number of serious concerns about the study:

The compilation of data seems to be very comprehensive at first sight, but when you dive into the database it is rather limited and some ice extents for a whole ice sheet is based on very few data points/models. This is problematic in particular when a lot of studies have not been included in the compilation from for e.g. SIS and GIS (see reviews by Sejrup et al., 2000; Houmark-Nielsen et al., 2003; Mangerud 2011; Funder et al., 1989, 2011; Vasskog et al 2015; Bierman et al., 2016 and many more). I am not expert in the other areas of the NH and cannot comment in detail on the data compilation from these regions, but the number of data points is often very low. If all the data is not included it should be described how the data is selected for this compilation. Other data points are misplaced c.f. Strunk et al 2017 modelled ice extent based on Al/Be measurements in bedrock in West Greenland – not Baffin Bay! The database clearly needs more attention – more data should be included.

One of the most serious flaws concern the way the best-estimate ice margins are produced. An example is the Early Matuyama time slice (1.78-2.6 Ma) which is based on three reconstructions (Marks, Kleman, Rea) and one data point for EIS (Montelli), but nothing from Asia. The data from Marks is 1.4 Ma i.e. not relevant for this time slice and this is also acknowledged in the suppl. text but why is it then on the map? - this is misleading. Also, in Asia many ice caps are present on the best-estimate but there is absolutely no data to support it. In the supplementary text it is explained that the best estimate for MIS4 is used for the Early Matuyama time slice “due to the similarity of d18O record”. This is really misleading as it is claimed in the main text that the new reconstructions are based on a synthesis of data and models (e.g. line 19-20, line 53-55, line 82-98). I have not seen any reconstructions of past ice extents based on d18O records alone, and this should be clearly described and discussed in the main text how this is done and what criteria is used to for example use MIS 4 and not MIS2 or MIS6 for the Early Matuyama.

In other time slices (e.g. MIS16, MIS12) younger more well-defined ice extents are also partly used for best-estimate. Actually for most pre-MIS6 glaciations the ice extent is to some extent based on

MIS6 or MIS4 reconstructions due to lack of actual data and/or models. In other instances, they are just prescribed by the authors as with the GIS extent without any explanation why it is on mid-shelf or outer shelf position. This clearly gives a bias, because the younger glaciations often have large ice extents (MIS4 or MIS6) and when they are used as best-estimate for older glaciations this might overestimate the ice extents in the older time slices.

When ice sheet reconstructions like the ones presented here are used to explain “ice sheet asymmetry” that result from “complex ice-ocean atmosphere interactions” I really get worried. All of the time slices older than MIS6 are very uncertain and it is extremely difficult to conclude anything about asymmetry etc when there is no data to support it. So, by using MIS4 or MIS 6 ice reconstruction as best-estimate for older glaciation because of lack of data very little can be basically be inferred for the older glaciations. You cannot argue that the asymmetry observed with the last glaciation is similar during older glaciations when the outline of older glaciations is based on the younger ice extents. That is circular argumentation.

In the section “Variations in ice-sheet extent” a number of bold statements are presented. For example, it is argued that GIS and CIS varied little between the reconstructed time slices. This cannot be concluded because there are very little or no data from Greenland in most periods and the ice extent is simply inferred to either be at the mid-shelf or present coastline (see suppl info). Obviously, there is little variation when the authors have used the same ice extent in every time slice in the absence of data!

The above-mentioned bias is also reflected in the intensity map (Figure 4). When all the maps are stacked in the intensity plot it produces a map that automatically looks like the MIS6 and MIS4 extents i.e. very large ice extent. It also shows that large areas where ice covered several times during the Quaternary because the MIS6 and MIS4 ice extents have been used in many time slices. This is an artefact of the way the maps have been made by inferring best estimates based on MIS4 or MIS6 extents. This might be ok but it is not supported by data and this is not discussed in the main text and is a major weakness.

In summary, the manuscript in its present state reflect an incomplete review rather than providing new insights into the NH ice sheet evolution the last 2.6 Ma. As the data is presented now it is difficult to see what new insights have been gained by this compilation. The asynchronous development of NH ice sheets has been documented previously for the last glacial cycle and the new compilation might show the same pattern for older glaciations although the data density is probably too low to conclude that. This is even acknowledged in the authors (line 115-119). Accordingly, I find the manuscript immature for publication. However, it has big potential and I urge the authors to 1)

provide a more complete database, and 2) re-assess the way ice margin extents are produced and used in discussing NH ice sheet evolution during the Quaternary.

Reviewer #1

The authors present a temporal succession of the Northern Hemisphere ice-sheets coverage that consists of 17 discrete time snapshots spanning the last ~1.0 Ma. Each time slice stems from the synthesis, digitization and georeferentiation of published empirical evidences (data points and outlines) and numerical models. The authors adopt a methodological framework where decisions are made according to a ranking scheme that generally gives priority to data over models. Each reconstructed temporal snapshot is accompanied by a minimum, maximum and best estimate and is ranked from low to high on the basis of a robustness score. The latter depends on the availability and agreement between empirical data and models and is characterized, to some degree, by unavoidable subjectiveness.

The results show that NH ice sheets were characterized by significant spatial differences and asymmetry throughout the Quaternary. The reconstructed global ice-sheets volumes are generally consistent with the eustatic curve based on delta-18O curve but are significantly smaller during MIS 3, as supported also by sedimentological observations (Pico et al., 2016). The authors point out that there is a general mismatch between the ice sheets margins that are based on empirical evidences and the outcome of numerical models. The proposed methodology and its outcome could therefore add additional and necessary constraints to the numerical models.

This paper does not present novel data or model output, but a longed-for capitalization on existing empirical data and numerical models that, through a sound and convincing methodological framework, provides input for a large scientific community.

Several disciplines such as biogeography, paleoclimatology, paleoceanography and paleontology will benefit from this research. In particular, ice-sheets modellers will be stimulated to combine the reconstructed NH ice-sheets margins with paleo RSL indicators and use them as constraints in their algorithms. The most advanced ice-sheets models do currently account for glacial and hydro isostasy (GIA) and will soon include sediments isostasy. Therefore, the inclusion of realistic ice-sheets margins as constraints would definitely contribute to a full transition from 'ice-sheet models' towards actual 'Earth systems models'. Also, realistic NH ice-sheets margins are a necessary when the GIA-driven RSL changes across interglacial are to be evaluated and removed in order to constrain the Greenland and Antarctic Ice Sheets retreats (see all the recently published works on MIS 5e).

Overall, the manuscript is very well written, it accounts for a comprehensive review of existing data and models and is accompanied by clear and exhaustive Methods and Supplementary Notes. The latter allow for a straightforward reproduction of the presented results. The figures and the Extended data are clear.

We thank the reviewer for their positive and encouraging comments.

Comment/request for the authors:

1) Several models are considered in this work. Among these, some have been constructed by including available ice-sheets margins based on empirical data (see ICE-5G from Peltier (2009) and ANU from Lambeck et al. (2006)). Do the authors think that there could be a bias due to this sort of circular problem? The same holds for the ice sheet models that are forced

by the delta-18O curve (see de Boer et al., 2014). Can the authors mention in the Supp. Notes whether a model is based on existing empirical outlines?

We have added a note to each modelled outline in the Supplementary tables, describing how the model was constructed and if it uses existing empirical data and/or d18O records.

We have added a sentence on this to the Methods:

‘Some of the models included in our compilation have been constructed or calibrated using existing empirical data about the ice margins and/or benthic $\delta^{18}\text{O}$ stack, as described in the data tables of the Supplementary Notes. Although such model outputs could produce some circular reasoning, we note that our best-estimate reconstructions are rarely informed by modelled outlines.’

Typos and errors:

Main text

1. Missing year in reference 36, Spratt et al. (line 355) Amended
2. Misspelled name of Lisiecki in Figure 2b, legend (change Liesicki) Amended

Supp. Notes, References:

1. Missing Year in Eccleshall et al., (line 1268) Amended
2. Missing Year in Gozhik et al., (line 1293) Amended
3. Missing Title in Kleman et al., (line 1363) Amended
4. Missing Year in Montelli et al., (line 1428) Amended
5. Missing Title in Nikolskiy et al., (line 1439) Amended
6. Missing Year in Rea et al., (line 1469) Amended

Reviewer #2 (Richard Gyllencreutz)

This manuscript presents ice sheet reconstructions for major paleo-ice sheets in the Northern hemisphere through the Quaternary. The times decided for reconstruction are pragmatically chosen based on research interest and data availability, and represent every 5 ka from LGM to 45 ka + MIS 5abcd, 6, 8, 10, 12, 16, 20-24, and for the paleomagnetic chrons Early Matuyama and Gauss. The timings for the MIS-time slices are based on Railsback et al. (2015), which is very sensible to avoid misunderstandings of MIS notation and/or timing. The 17 reconstructions are based on a comprehensive synthesis of published data from empirical ice sheet-wide reconstructions, empirical point data, and where empirical data is lacking, from numerical model results. The reconstructions are supplied with maximum, best-estimate, and minimum extents and a robustness score, which although somewhat subjective greatly facilitates assessment and use of them.

The manuscript is very well written, concise and has a clear structure. The main manuscript is supported by a separate Methods chapter, and Extended Data with larger and more detailed figures to be used together with Supplementary Notes containing all relevant metadata about the data used and the rationale behind the interpretations. The manuscript contains 4 figures, which are all relevant and well designed to be as small as possible while clearly showing what they are aimed for. Some details about the figures are given below.

The main conclusions from their analysis of the new reconstructions are:

- There is large ice sheet asymmetry between different glacials
- The Eurasian Ice Sheet showed the largest asymmetry
- The Laurentide Ice Sheet was much reduced during MIS 3, implying 30-40 m higher sea levels at that time than previous estimates, which is supported by recent MIS 3 sea level data from Pico et al. (2016).

The conclusions are reasonable and they are well supported by the data presented. However, it is the reconstructions (maps in Fig. 1 and Extended Data) that constitute the main contribution of this manuscript.

This manuscript presents, through a consistent methodology, the most comprehensive and accurate synthesis of NH Quaternary Glaciation extent to date, and will be of great value to many fields of research, including paleoclimatology, paleoglaciology, numerical modelling of ice sheets and climate, archaeology, and paleontology.

We thank the reviewer for their positive and constructive review.

Specific comments

Main Manuscript

(line numbers refer to PDF version. Word-document was numbered differently)

1) Line 157: ‘... good fit with previously published global sea-level curves ...’ It may be relevant to mention that the presented LGM reconstruction matches the result of ~100 m sea level equivalent for the northern Hemisphere by Simms et al., 2019 (QSR, <https://doi.org/10.1016/j.quascirev.2018.12.018>). Simms et al. study the mis-match between ice sheet and paleo-sea-level reconstructions, and conclude that ~15 m of sea level appears ‘missing’ in the LGM reconstructions.

To address this point, we have added the following sentences to the section on sea-level equivalent ice volume (Line 203):

‘The sea-level equivalent volume of our LGM reconstruction, which is based mainly on an existing compilation of empirical evidence (Ehlers *et al.*, 2011) (Supplementary Notes), closely matches the *c.* 100 m sea-level equivalent for the NH ice sheets that has been estimated by other studies (Simms *et al.*, 2019). The discrepancy between this estimate and the *c.* 130 m of sea-level equivalent that is suggested by the benthic $\delta^{18}\text{O}$ stack (Fig. 2b) may be the result of potential inadequacies of current models in estimating glacial isostatic adjustments (Simms *et al.*, 2019) as well as the exclusion of Southern Hemisphere ice masses from our study.’

2) Line 162 (see also Fig 2b below): ‘These cumulative ice volumes do not include Southern Hemisphere ice masses and assume that the NH ice sheets reached their maximum extent around the same time and, therefore, should be viewed as a maximum sea-level lowering’.

This is confusing for two reasons: 1) the wording and 2) the assumption.

1) Because the amount of lowering is minimum when the sea level is at maximum (i.e. at its shallowest).

2) If the NH ice sheets DID NOT reach their maximum at the same time, the amount of sea level lowering would be smaller (i.e. the sea level would be higher), making the reconstruction a MAXIMUM amount of lowering. However, if the SH ice sheets WERE included, the amount of sea level lowering would be larger (i.e. the sea level would be lower), making the reconstruction a MINIMUM amount of lowering. Thus, the reconstructed sea level is neither a minimum nor maximum, just approximate.

As there is no way around this problem but to include SH ice sheets and/or make synchronous time slice maps, I suggest that the end of this sentence is rewritten as something like ‘and, therefore, the sea levels are plotted at the times of lowest global sea level, and represent the maximum amount of sea level lowering from NH glaciation’.

We have reworded this sentence as below. We have moved the discussion of the exclusion of Southern Hemisphere ice masses to later sentences in this paragraph.

‘These cumulative ice volumes assume that the NH ice sheets reached their maximum extent at the same time and, therefore, are plotted at the times of lowest global sea level, and represent the maximum amount of sea level lowering from NH glaciation.’

3) Figure 2b (see also Line 162 above):

‘Because our cumulative ice volumes assume that the NH ice sheets reached their maximum extent around the same time, our sea-level-equivalent estimates for the full-glacial periods of MIS 2, 4, 6, 8, 10, 12, 16 and 20-24 are plotted at the coldest point (lowest global sea level) within each of these time-slices’.

First, ‘around the same time’ should be ‘at the same time’.

Second, here they use the same statement as on Line 162 (simultaneous NH ice sheet maximum in the reconstruction), but use it to explain where on the timescale the calculated sea levels are plotted. This is reasonable, and is also stated on Line 170-173 in Methods.

We have reworded this sentence to state ‘at’ the same time, rather than ‘around’ in both of these instances.

Figures

Fig. 1a: the symbol for ‘Outline from empirical data’ could be made multi-colored (e.g. divided in 3 fields with orange, light blue and light green) to be clearer.

We have changed this symbol to be multi-coloured, as suggested.

Fig. 2: All labels should be slightly bigger, so the figure could be printed smaller.

Because *Nature Communications* articles use a two-column format, figures are either full-page or half-page in width. We believe that half-page width would be insufficient to show clearly the information contained within Figure 2. We therefore prefer to keep Figure 2 as full-page width.

Note that we reduce the size of Figure 4 to half-page width, as suggested by Reviewer 2.

2a) The x-axis labels would be easier to connect to the correct bar if they were moved slightly to the left (so they point towards their respective bar instead of being centered under it).

2b: Please add minor tick marks for every 10 ka on the lower x-axis.

We have changed the orientation of the labels to be easier to connect to the correct bar, but prefer to keep the x-axis labels in the centre of each bar

Fig. 3: All labels should be slightly bigger, so the figure could be printed smaller.

Because *Nature Communications* articles use a two-column format, figures are either full-page or half-page in width. We believe that half-page width would be insufficient to show clearly the information contained within Figure 3. We therefore prefer to keep Figure 3 as full-page width.

Note that we reduce the size of Figure 4 to half-page width, as suggested by Reviewer 2.

Caption: The ETOPO1 dataset should be cited as ‘Amante, C. and B.W. Eakins, 2009. ETOPO1 1 Arc-Minute Global Relief Model: Procedures, Data Sources and Analysis. NOAA Technical Memorandum NESDIS NGDC-24. National Geophysical Data Center, NOAA. doi:10.7289/V5C8276M”. Please put this in the reference list, and refer to it as Amante et al. (2009) in the caption.

We have added this reference to the caption and references list.

Fig. 4: The intensity map is a great idea. However, I don’t see the point of translating the number of glaciations per area to a ‘glaciation intensity value’ of 0 to 1 with a smooth colour scale, as the glaciation intensity on the map is not continuous. It would be clearer if the legend showed discrete boxes of the corresponding colours from the map associated with the number of glaciations covering each area from 0 to 8 (or >8? The text says >8 but that’s not visible in the map).

We have changed the legend of this map to show the discrete number of glaciations per area, instead of a continuous glaciation intensity value.

We have changed the first paragraph of the section on landscape evolution to reflect these changes. It now reads:

‘Combining our best-estimate reconstructions for the last c.1 Ma (Fig. 4) shows the number of times that each region was covered by ice during the Mid- and Late Pleistocene time-slices sampled in this study. To account for the different lengths of these time-slices, only the largest reconstruction within MIS 3 (which spans time-slices 30, 35, 40 and 45 ka) and within MIS 5 (which spans time-slices MIS 5a–d) was used. Areas that were ice-covered during the two oldest, Late Pliocene to Early Pleistocene, time-slices, the early Matuyama (1.78–2.6 Ma) and late Gauss (2.6–3.6 Ma) magnetic chrons, were not included because they span such broad time periods. Although areas could have been ice-covered during additional glaciations, this map provides a useful conceptual framework to interpret the landscape evolution of the NH.’

This map has large enough labels for the figure to be made smaller, but then the legend needs to be larger (could be placed over southern Eurasia instead).

We have reduced the size of Figure 4 to half-page width and changed the size of the text and legend accordingly.

Methods

The referencing of the methods chapter is based on numbers in the reference list (not alphabetical), but references are not numbered in the text. Please make this coherent.

Thank you for pointing this out. We have changed the referencing of the methods chapter to be alphabetical.

The robustness scores are clearly defined and well justified, and are strengthening the manuscript. For the ice sheet outline digitization, several assumptions and base postulates are given. All are reasonable and necessary. I comment on the most, in my view, important ones below:

- locations of ice free-indicating points are not plotted, but were used to inform the reconstructions and described in the Supplementary Notes. (Line 41-44). This is a necessary decision, because most ice-free indications are redundant (only ice-free points close to other points overridden by ice are relevant), and plots would be cluttered if all were included.

- ‘the raw data maps [...] should not be used for local-scale studies or as a substitute for the original source data’ (Line 49-50).

This ensures that original publications get credit and discourages over-interpretation of the data on a local scale; this work regards ice-sheet wide extent.

- only ice masses > 50 000 km² were considered (Line 60-62). This is a reasonable cut-off for a hemispheric-scope reconstruction, i.e. excluding ice caps smaller than about the size of Denmark.

- ‘For regions and/or time-slices where empirical and modelled data are not available, a feasible ice-sheet extent was derived using the ice-sheet configuration from another time-slice that has a similar value in the global d18O record (Spratt and Lisiecki, 2016) (Supplementary Notes). It should be noted that, in these cases, estimates of the eustatic sea-level equivalent represented by the cumulative volume of the ice sheets are not fully independent.’ (Line 68-73). This is a clever and relatively transparent way of filling data gaps, although it makes calculations of sea-level contributions from those ice caps circular

and forces the sea level contributions in the direction to fit the Spratt and Lisiecki (2016) curve better. The authors are aware of this.

No response required.

Line 53-55: Please include a reference to Hughes et al. (2016) to acknowledge that their ice sheet reconstruction was the first to provide maximum, minimum and best-estimate lines for ice sheet extent. This was an important conceptual improvement in the usefulness (especially for modellers and for identification of areas/times where data is lacking) and credibility assessment of ice sheet reconstructions, which should be acknowledged.

We have added the following sentences to the Methods:

‘This approach builds upon that of Hughes, A. *et al.* (2016), whose reconstructions of ice-sheet extent were the first to use maximum and minimum limits to represent uncertainty. The use of maximum, minimum and best-estimate reconstructions in our study provides a visual indicator of uncertainty and identifies regions and time-slices where future work should be directed.’

We have also added a sentence on this to the main text to further validate our approach. This reads:

‘The use of max-min bounds has been used previously to illustrate uncertainty in the past extent of ice masses (Hughes, A. *et al.*, 2016).’

Line 167-168: ‘For each time-slice, the volume for the best, minimum, maximum area estimates were translated into global sea-level change by dividing the ice-sheet volume by the area of the world’s ocean, i.e., ~362 million km²’.

The authors are thus not compensating for the different ice and seawater densities of 917 and 1028 kg m³ in calculating their sea-level equivalents (SLE). The ice-to-water density ratio is 0.892, so neglecting this gives only 89.2% of the amount of sea level lowering (or equivalently, the sea level should be lowered by another 12.1%). Neither do they state whether they assume that sea water replaces ice where ice was grounded on the sea floor, which is important because buoyancy will cause margin ice sheet parts to not contribute with their full volume to the sea level change. This effect is minor for shallow water depths (e.g. about 1 m SLE for the Barents-Kara ice sheet at LGM at 210 m mean water depth; Hughes et al., (2016)), but may be important for the proposed grounded ice in the deep Arctic Ocean for MIS 4, 5b, 5d, 6, 12, 16, 20-24, Early Matuyama, and Late Gauss.

Our decision not to compensate for the different densities of ice and seawater is offset, to some degree, by our assumption that the NH ice sheets reached their maximum extent at the same time, which was most probably not the case. We have added some sentences about this to the main text:

‘These cumulative ice volumes assume that the NH ice sheets reached their maximum extent at the same time and, therefore, are plotted at the times of lowest global sea level, and represent the maximum amount of sea level lowering from NH glaciation. This assumption is compensated for, at least in part, by the fact that we do not account for the different densities of ice and seawater, which would produce an additional sea-level lowering of around 12%.’

We do not correct for sea water replacing grounded ice because of uncertainties about long-term bathymetry and ice thickness.’

Supplementary Notes

The supplementary notes are to be read together with the figures in Extended Data.

1) There are several occasions of statements like ‘Due to uncertainty [a certain extent] is used for [a certain part of an ice sheet]’ on Line 133, 148, 188, 238, 358, 421, 554, 610, 689, 694, 700, 703, 753, 763, 759, 933, 940, 1006, and 1077. What is uncertain is only specified in 3 of these (610, 763 and 940).

Please specify wherein the uncertainty lies for all these occasions, as they are the rationale for using some assumed ice extent in these times/locations. In most cases this is probably because no empirical data exist for this location at this time, but it should be explicitly stated if that is the case.

Yes, this is because there are no empirical data for these location at these times. We have changed the wording in these instances to explicitly state this.

2) MIS 6 Table: Graphical comment - the blue line in the legend for ‘Colleoni et al., 2016’ is misplaced and should be moved slightly to the left.

We have amended this.

3) Line 721-723: ‘... but the three ice advances that are proposed between 0.2 and 0.75 Ma most likely occurred in MIS 6, 12 and 16’. Why most likely at those times? I suppose this is based on the fact that MIS 6, 12, and 16 are the most extensive glaciations in this time span according to the global sea level record of Spratt & Lisiecki (2016), but the reason should be written.

Yes, this is because MIS 6, 12 and 16 are the most extensive glaciations in this time span according to the global sea level record. We have added some text to the Supplementary to clarify this.

4) Early Matuyama Table and Explanatory text Line 1014, and Late Gauss Table: Data for the LIS is used with reference to ‘Barendregt pers. comm.’

As shown in Extended Data Figure 8c and 8d, the entire North American ice sheet complex, except for a small patch at the southernmost extension of the LIS in Early Matuyama, is based on ‘Barendregt pers. comm’. It would not be acceptable to base an interpretation of a continent-wide glaciation on unpublished and undisclosed data - how does the reader know the data even exists?

However, the Barendregt data is actually not used (or at least seems redundant) - the reconstruction for Early Matuyama is (as far as one can discern in the printed figures) based entirely on the maximum MIS 6 outline and Balco & Rovey (2010), and for Late Gauss is based on Kleman et al. (2010) and Kaufman et al. (2011).

Either make a case that the Barendregt data is important, and preferably include it as a table, or omit it from the manuscript. You can mention in the supplementary text that your interpretation is supported by the unpublished Barendregt data, but not that it is based on it.

We contacted Rene Barendregt about how to best reference this work. These figures are from a 2014 conference poster and are modified from Barendregt and Duk-Rodkin (2011). For completeness, we have added the published outlines of Barendregt and Duk-Rodkin (2011) to our compilation for the early Matuyama and late Gauss time-slices, as well as the versions from the 2014 poster.

The schematic outline of Barendregt et al. (2014) was drawn to cover the accompanying empirical (palaeomagnetic) data points, and is therefore a minimum reconstruction. We use this outline for our minimum early Matuyama reconstruction. We have added a note to the Supplementary to explain why we do not use this outline in the best-estimate. We do not use Barendregt's outline for the late Gauss chron because our approach is to use one of a series of 'templates' for the CIS (see Methods).

Supplementary Notes References

Line 318: Mangerud (2011) - is this Mangerud et al. (2011)?

Yes, we have amended this.

Line 1363: Kleman et al., 2008 - the title of the paper 'Patterns of Quaternary ice sheet erosion and deposition in Fennoscandia and a theoretical framework for explanation' is missing.

Amended

Reviewer #3

The manuscript ‘configuration of Northern Hemisphere ice sheets through the Quaternary’ presents 17 pre-LGM ice sheet reconstructions based on a compilation of existing data and models. In addition, they calculate ice volume and discuss various mechanisms to explain the apparent ice sheet asymmetry in build-up and decay of sheets during the Quaternary. It has clearly been a large effort to compile the data. The research idea is excellent and the potential outcome of this exercise could potentially reveal something ground-breaking about the NH ice sheet evolution in the Quaternary that would be suitable for Nature Communications.

However, in its present form I cannot recommend publication as there are a number of serious concerns about the study:

1) The compilation of data seems to be very comprehensive at first sight, but when you dive into the database it is rather limited and some ice extents for a whole ice sheet is based on very few data points/models. This is problematic in particular when a lot of studies have not been included in the compilation from for e.g. SIS and GIS (see reviews by Sejrup et al., 2000; Houmark-Nielsen et al., 2003; Mangerud 2011; Funder et al., 1989, 2011; Vasskog et al 2015; Bierman et al., 2016 and many more). I am not expert in the other areas of the NH and cannot comment in detail on the data compilation from these regions, but the number of data points is often very low. If all the data is not included it should be described how the data is selected for this compilation.

A) How data are selected for this compilation

We have added a paragraph to the main text that describes how data were selected for this compilation:

‘Empirically derived and numerically modelled outlines of ice-sheet extent were the primary targets of our literature search for evidence for NH ice sheets. Although it is beyond the scope of this study to review all marine-sedimentological evidence for ice-sheet growth and decay (e.g. ice-rafted debris), evidence derived from sedimentological and stratigraphic investigations was incorporated into our reconstructions (Supplementary Notes). These data types were specifically targeted for older time-slices for which published ice-sheet outlines are scarce. Our compilation does not included data that have been superseded by the same author(s) or are located well inside an established ice margin (Methods). With the exception of the comparatively warm periods of 45 ka, MIS 5a and 5c, for which we aim to capture the peak warmth, our reconstructions aim to show the maximum ice-sheet extent within each time-slice (Methods). This is particularly important to note for the oldest time-slices (i.e. early Matuyama and late Gauss magnetic chrons), which span broad periods of time that included significant fluctuations in ice-sheet extent (e.g. Laberg et al., 2013).’

We have also added detail on how data were selected to the Methods, which reads:

‘Our literature search was based on the following general principles:

- We are interested in mapping the changing ice-marginal position of the ice sheets and therefore did not include data points that are located well inside a suggested ice margin.
- In cases where the same author(s) have published multiple reconstructions for the same area, we used the most recent hypothesised ice-sheet extent. An example is that we do

not include the data of Houmark-Nielsen and Kjær (2003) because these have informed and been superseded by later work from the same author (Houmark-Nielsen, 2010).

- When using ice-sheet outlines that are derived from a synthesis of previously published empirical evidence (e.g. Svendsen et al., 2004; Knies et al., 2009), we did not include all of the data points that informed the synthesised reconstruction.
- It is beyond the scope of this study to review all marine-sedimentological evidence for ice sheets (i.e. ice-rafted debris). Sedimentological and stratigraphic data (including marine seismic data) were used to supplement empirically derived and numerically modelled ice-sheet outlines and were particularly targeted for the oldest time-slices (early Matuyama and late Gauss palaeo-magnetic Chrons) where published ice-sheet outlines are scarce.'

B) Data recommended for inclusion

We have checked the studies that Reviewer 3 suggests we have omitted. Below are detailed comments on these:

- Sejrup et al., 2000.

This paper includes a glacial curve diagram that shows the suggested extent of the EIS in MIS 4, 6, 8, 10, 12 and 20-24 as inferred from seismic data. We have added these data points to these time-slices.

- Houmark-Nielsen et al., 2003.

We cannot find this work. Perhaps Reviewer 3 means Houmark-Nielsen and Kjaer, 2003? This was not included in our raw data as it has been superseded by more recent work by the same author (Houmark-Nielsen, 2010). We use the reconstructions of Houmark-Nielsen (2010) extensively, as noted in the Supplementary.

We include a new paragraph in the main text and Methods that clarifies how our literature search was performed.

- Mangerud, 2011.

We cannot find this work. Perhaps Reviewer 3 means Mangerud et al., 2011?

We use the reconstructions of Mangerud et al. (2011) extensively, as noted in the Supplementary (in 35 ka, MIS 4, 5a, 5b, 5c and 5d).

- Funder et al., 1989.

We cannot find this work. Perhaps Reviewer 3 means Funder, 1989?

Funder, 1989 includes a glacial curve diagram that suggests an ice sheet existed in NW Greenland during MIS 6 and MIS 5d (114 ka).

We have added these data points to our raw data maps for MIS 5d and 6.

- Funder et al., 2011.

Figure 50.1 of this work suggests that the Hellefisk moraines of West Greenland are MIS6 in age, as proposed by Kelly (1985) from correlation with weathering boundaries.

However, the authors state in the text that their age is an 'open question.'

Other more recent studies attribute these moraines to the LGM or Younger Dryas (e.g. Roberts et al., 2009).

The GIS is now generally acknowledged to have extended to the shelf break beyond Greenland during the LGM (e.g. Evans et al., 2009; Dowdeswell et al., 2010; Ó Cofaigh et al., 2013) and MIS 6 (e.g. Nielsen and Kuijpers, 2013; Strunk et al., 2017).

We therefore prefer not to include this data point for MIS 6.

- Vasskog et al., 2015

This paper provides a summary of the GIS during the last glacial cycle.

There is a short section on MIS 3-5d, and the rest of the paper focuses on the last interglacial, LGM and Holocene (which are not reconstructed in our study).

The section on MIS 3-5d does not contain any information about the position of the GIS margin during these time-slices, apart from that there is IRD evidence for periods of ice-sheet expansion (Stein et al., 1996).

We have added the data points of Stein et al., 1996 to show that the GIS reached at least a coastline position at 30 ka, 35 ka, MIS 4, 5d and 6.

- Bierman et al., 2016.

This paper includes IRD evidence for the GIS being marine-terminating in SE Greenland during MIS20-24 and the early Matuyama. We have added data points to our raw data maps for these time-slices. Note the section in the Methods chapter that describes how IRD evidence for ice sheets is plotted.

C) Additional literature search

In light of the comments made by Reviewer 3, we performed a thorough literature search for additional data. This search yielded a further: 9 empirically derived ice-sheet outlines; 24 data points representing dated sediments, seismic data, IRD or glacial curve diagrams; 1 modelled outline; and 15 additional references. These additional data are mainly marine seismic or IRD data and they relate mainly to our oldest time-slices.

We have added these additional data to the tables and text in the Supplementary. The addition of these data resulted in some modifications to our reconstructions, which are incorporated in our revised figures.

Other data points are misplaced c.f. Strunk et al 2017 modelled ice extent based on Al/Be measurements in bedrock in West Greenland - not Baffin Bay!

The database clearly needs more attention - more data should be included.

Previously, our red data points in the raw data maps were placed at the location that ice was inferred to have reached, which is not always the location that the data were collected from. We have changed this approach to make it more robust. In our new approach, data points are placed at the locations where the data were collected, and arrows show where the ice was inferred to have reached. We have therefore changed the location of the Strunk data points to close to the coastline of West Greenland, and added arrows showing that the glacial curve figure in this paper suggests that ice reached the shelf break in Baffin Bay.

We have added an explanation of this approach to our Methods section, which reads:

‘Ice-marginal positions that are inferred from studies of ice-rafted debris (IRD) in sediment cores were included as point-source data. In these cases, the data point (red circle) was placed at the position that the core was taken, and an arrow shows the location that the ice was inferred to have reached. Unless a glacial curve diagram is also included, the presence of IRD in a marine sediment core is taken to indicate that the ice sheet reached the present-day coastline.’

2) One of the most serious flaws concern the way the best-estimate ice margins are produced. An example is the Early Matuyama time slice (1.78-2.6 Ma) which is based on three reconstructions (Marks, Kleman, Rea) and one data point for EIS (Montelli), but nothing from Asia.

A) Lack of data from some regions (e.g. NE Asia) and time-slices

Our additional literature search resulted in the incorporation of a further 9 empirically derived outlines and 24 data points. No empirical data are used in our Early Matuyama reconstruction for NE Asia because we have not found any data about the location of ice in this area during this time-slice in the published literature. This paucity of empirical data is indicated by a robustness score of 0 marked on the relevant map for NE Asia in the Extended Data.

B) Transparency about the robustness of our reconstructions

We believe that our paper is transparent about the level of uncertainty in our reconstructions. To use the example of ice in NE Asia during the early Matuyama, the high level of uncertainty is illustrated in our reconstructions in three main ways:

1) By our min-max bounds. Here, the max reconstruction is the maximum Quaternary extent (of Glushkova, 2011 and Barr and Clark, 2012), and the min reconstruction is no ice sheet.

2) By the robustness score, which is 0 (no data).

3) By the raw data maps. It is apparent from the raw data map for the early Matuyama Chron (Extended Data), which, for ease of comparison, is located alongside the reconstruction for this time-slice, that there are no empirical or modelled data points for this region.

Despite the uncertainty about ice in NE Asia during this time-slice, we believe that our reconstruction of ice sheets during the early Matuyama Chron is valuable because a significant amount of empirical data (see table 16 of the Supplementary) exist for the other ice sheets during this time.

- In light of these comments by Reviewer 3, we have **added the individual and overall robustness scores to the maps in Fig. 1**, to make the level of uncertainty in each region/ time-slice more obvious to the reader.

C) Transparency about the approach used to produce reconstructions in regions/ time-slices where there are limited/ no data

We agree with Reviewer 3 that the main text should include more detail about the approach that was taken to produce reconstructions in regions/ time-slices where there are limited/no data. We have added a new paragraph and other additional text to the main manuscript to explain our approach in these cases. This is shown below (old text in green, new text in blue):

‘Following the compilation of the available evidence, we then produce new hypotheses relating to ice-sheet extent that span the Quaternary (Fig. 1d–u). For each time-slice we capture uncertainty by defining a maximum and minimum limit (Fig. 1a and b) and provide a ‘best-estimate’ hypothesis (Fig. 1d–u; Supplementary Notes). Max-min bounds have been used previously to illustrate uncertainty in the past extent of ice masses¹¹. Our

best-estimate reconstructions are scored from low to high confidence using a robustness score (Fig. 1d-u) that is based on the availability and agreement between modelled and empirical constraints for that time-slice. Some of our reconstructions are well-constrained by empirical data; for example, we present a reconstruction of the maximum extent of the NH ice sheets during MIS 6 that is constrained mainly by empirical data (Fig. 1a and b). However, comparatively few data exist about ice-sheet extent during older time-slices, interstadial periods (e.g. 45 ka, MIS 5a and 5c), and glacial periods such as MIS 8 and 10 that occurred between glaciations of larger extent. There is also spatial variability in the distribution of empirical data, with information about past ice sheets particularly lacking from NE Asia (Extended Data).

In regions where there are few or no existing data for a time-slice, we use a reconstruction from another time-slice that has a similar value in the benthic $\delta^{18}\text{O}$ stack (Lisiecki and Raymo, 2005) to construct a feasible ice-sheet margin. Because the best-constrained ice extents are often the youngest, some of our older reconstructions are based, in part, on ice-sheet extents from younger time-slices. For example, the best-estimate LIS during MIS 12 incorporates the best-estimate reconstruction for MIS 6 where empirical data (Balco and Rovey, 2010) are absent (Supplementary Notes). To avoid unnecessary complexity in regions where empirically derived reconstructions are scarce, ice-sheet ‘templates’ were used for the North American Cordilleran, Greenland, Iceland and NE Asia (see Methods). For example, three ice-mass configurations are used for NE Asia: 1) the Quaternary maximum (Glushkova, 2011; Barr and Clark, 2012; 2) the LGM (Barr and Clark, 2012); and 3) no ice sheet. The use of templates and ice-sheet extents from other time-slices is necessary to fill the current gaps in our knowledge of Quaternary ice-sheet extent, and is an improvement on methods that use the LGM as input for all Quaternary glaciations.

In total, we reconstruct a maximum, minimum and best-estimate NH ice-sheet extent for 17 separate time-slices prior to the LGM, and a best-estimate for the comparatively well-constrained LGM^{8,11,13,14}. Although our best-estimate reconstructions are informed by some subjective decisions, they provide hypothesised first-order reconstructions of NH ice sheets through the Quaternary that are based on available empirical evidence.’

The data from Marks is 1.4 Ma i.e. not relevant for this time slice and this is also acknowledged in the suppl. Text but why is it then on the map? - this is misleading.

We have removed the data of Marks (2011) from the raw data map for the early Matuyama Chron. We keep the discussion of these data in the Supplementary because of uncertainty in dating older sediments.

Also, in Asia many ice caps are present on the best-estimate but there is absolutely no data to support it.

We are transparent about the lack of data to support these reconstructions (including giving the reconstruction in this area a robustness score of 0: no data). We have added a paragraph (see above) that explains our approach in regions/ time-slices where no data are available.

In the supplementary text it is explained that the best estimate for MIS4 is used for the Early Matuyama time slice ‘due to the similarity of $\delta^{18}\text{O}$ record’. This is really misleading as it is claimed in the main text that the new reconstructions are based on a synthesis of data and models (e.g. line 19-20, line 53-55, line 82-98). I have not seen any reconstructions of past

ice extents based on d18O records alone, and this should be clearly described and discussed in the main text how this is done and what criteria is used to for example use MIS 4 and not MIS2 or MIS6 for the Early Matuyama.

We agree with Reviewer 3 that this approach should be made obvious in the main text, as to not be misleading. We have added a paragraph to the main text (see above) that details how and why we use the d18O record to inform reconstructions in cases where there are limited/no data.

In other time slices (e.g. MIS16, MIS12) younger more well-defined ice extents are also partly used for best-estimate. Actually, for most pre-MIS6 glaciations the ice extent is to some extent based on MIS6 or MIS4 reconstructions due to lack of actual data and/or models.

Our new paragraph in the main text (see above) includes some sentences about this point, which reads:

‘Because the best-constrained ice extents are often the youngest, some of our older reconstructions are based, in part, on ice-sheet extents from younger time-slices. For example, the best-estimate LIS during MIS 12 incorporates the best-estimate reconstruction for MIS 6 where empirical data (Balco and Rovey, 2010) are absent (Supplementary Notes).’

We note, however, that there is terrestrial and marine evidence for older glaciations reaching a similar extent to younger glaciations, as discussed below.

In other instances, they are just prescribed by the authors as with the GIS extent without any explanation why it is on mid-shelf or outer shelf position. This clearly gives a bias, because the younger glaciations often have large ice extents (MIS4 or MIS6) and when they are used as best-estimate for older glaciations this might overestimate the ice extents in the older time slices.

The marine-terminating portions of many ice sheets (including Greenland, Iceland, eastern LIS, western CIS) *are* known to have reached the same position during older Quaternary glaciations as during younger glaciations – the continental shelf break. The evidence for this is in the form of large glacial-sedimentary depocentres (trough-mouth fans) that have built up on the continental slope through the Quaternary (e.g. Li et al., 2011; Laberg et al., 2013; Hoffman et al., 2016).

We note that there is a comment about shelf-break glaciation during multiple Quaternary glacial periods in the section on landscape evolution, which reads:
‘Locations where ice sheets reached the continental shelf break during multiple Quaternary glaciations (e.g. Norwegian, Greenland, northern and eastern Canadian, and Barents-Kara Sea margins) are also key sites of glacial deposition, as indicated by major (up to 1 km-thick) glacial-sedimentary depocentres on the continental slope^{45,46}.’

We recognise that the *location of the shelf break* has changed because it has prograded seaward through the Quaternary. However, as we note in the Methods, we do/ can not account for changing margin architecture in our reconstructions.

We have added a significant amount of text (around 13 pages) to the Supplementary, which better explains the positions of the ice margins in the various time-slices. Examples include:

For MIS 5a:

‘Ice in Greenland and Iceland is shown at the present-day coastline because this is the mid-point between the maximum and minimum reconstructions.’

For MIS 6:

‘The depiction of the GIS at the shelf break is in agreement with work that has inferred extensive glaciation of East Greenland during MIS 6 (Funder et al., 1998; Nielsen and Kuijpers, 2013).’

For MIS20-24:

‘The reconstruction of the GIS at the shelf break during MIS 20–24 is in agreement with an increase in IRD at around 0.8 Ma (Bierman et al., 2016), and seismic evidence for multiple cross-shelf glaciations between 0.78 and 1.77 Ma (Laberg et al., 2013).’

For the late Gauss:

‘The GIS is shown at the shelf break in our best-estimate reconstruction. This is in agreement with empirical and modelling work that suggests that the GIS extended to the shelf break during the Late Pliocene to Early Pleistocene, between around 2.5 and 3 Ma (Solheim et al., 1998; Berger and Jokat, 2009; Nielsen and Kuijpers, 2013; Solgaard et al., 2011; Knutz et al., 2015; Bierman et al., 2016; Hoffman et al., 2016; Pérez et al., 2018). An expanded GIS during this time is also suggested from IRD records (e.g. Jansen et al., 2000; Bailey et al., 2012).’

3) When ice sheet reconstructions like the ones presented here are used to explain ‘ice sheet asymmetry’ that result from ‘complex ice-ocean atmosphere interactions’ I really get worried. All of the time slices older than MIS6 are very uncertain and it is extremely difficult to conclude anything about asymmetry etc when there is no data to support it. So, by using MIS4 or MIS 6 ice reconstruction as best-estimate for older glaciation because of lack of data very little can be basically be inferred for the older glaciations. You cannot argue that the asymmetry observed with the last glaciation is similar during older glaciations when the outline of older glaciations is based on the younger ice extents. That is circular argumentation.

We do not wish to make any inferences about ice-sheet asymmetry in older glaciations, *only* for the last glacial cycle (for precisely the reasons stated by Reviewer 3). Our only mention we make of asymmetry in older glaciations is the evidence derived from ice cores.

This was acknowledged in our original manuscript:

‘Although it is not currently possible to assess geological evidence for NH ice-sheet asynchronicity within older glacial periods, records of global dust flux derived from Antarctic ice cores show a pronounced double peak within many earlier glacial cycles...’

In response to this comment, we have changed the main text to remove any instances in which this could be ambiguous:

- We have added ‘during the last glacial cycle’ to a paragraph on ice-sheet asymmetry.
- We have changed a sentence in the abstract, which now reads:

‘Our hypothesised reconstructions illustrate pronounced ice-sheet asymmetry within the last glacial cycle (Marine Isotope Stage (MIS) 2–5d), and significant variations in ice-marginal positions between older glacial cycles.’

4) In the section ‘Variations in ice-sheet extent’ a number of bold statements are presented. For example, it is argued that GIS and CIS varied little between the reconstructed time slices. This cannot be concluded because there are very little or no data from Greenland in most periods and the ice extent is simply inferred to either be at the mid-shelf or present coastline (see suppl. info). Obviously, there is little variation when the authors have used the same ice extent in every time slice in the absence of data!

We acknowledge that there are limited data to determine the extent of the GIS and CIS in every time-slice but believe that the maximum and minimum limits constrain the variation to a relatively small amount compared to that of the LIS or EIS. In terms of the maximum limit, it is not possible for the GIS and the western margin of the CIS to expand beyond the continental shelf break. In terms of the minimum limit, the modern GS covers about 80% of the island during the present (warm) interglacial conditions. Therefore, the area of the GIS must have varied comparatively little between the reconstructed (cold) time-slices. We have rephrased these sentences to make our point clearer:

‘The Greenland Ice Sheet (GIS) and CIS have a comparatively small magnitude of variation in ice-sheet area between the reconstructed time-slices (Fig. 2a). Although some of our reconstructions are poorly constrained by empirical data, it is apparent that the relatively narrow continental shelf beyond Greenland and western Canada limits the maximum size that the GIS and CIS can attain.

5) The above-mentioned bias is also reflected in the intensity map (Figure 4). When all the maps are stacked in the intensity plot it produces a map that automatically looks like the MIS6 and MIS4 extents i.e. very large ice extent. It also shows that large areas where ice covered several times during the Quaternary because the MIS6 and MIS4 ice extents have been used in many time slices. This is an artefact of the way the maps have been made by inferring best estimates based on MIS4 or MIS6 extents. This might be ok, but it is not supported by data and this is not discussed in the main text and is a major weakness.

There *is* empirical evidence that the ice sheets in the older glaciations included in this figure (MIS 12, 16 and 20-24) reached as far south as they did during younger glaciations. This includes terrestrial evidence in Europe (Astakhov, 2016) and North America (Colgan, 1999; Balco and Rovey, 2010). In addition, the marine-terminating portions of many ice sheets (including Greenland, eastern LIS, western CIS) have been interpreted to have reached the continental shelf break during multiple pre-MIS 6 glaciations (e.g. Li et al., 2011; Batchelor et al., 2014; Montelli et al., 2017).

Our additional literature search identified several new empirical outlines and data points (see Supplementary Notes). The only best-estimate reconstructions that now rely on reconstructions from younger time-slices are the EIS in MIS 16 and the LIS in MIS 8, 10, 12 and 16. In the case of the LIS in MIS 12 and 16, the *most southern* limit of this ice sheet is actually the part that is supported by the empirical data (Colgan, 1999; Balco and Rovey, 2010).

We have added some sentences to this section that acknowledge the point made by the Reviewer. It reads:

‘Although some of the older ice-sheet reconstructions that informed Figure 4 are based, in part, on ice-sheet extents from younger time-slices, we note that there is empirical evidence

for NH ice sheets reaching a similar southerly position between around 0.4 and 1 Ma (MIS 12, 16 and 20-24) as during younger glaciations (Balco and Rovey, 2010; Andriashek and Barendregt, 2017; Extended Data).’

In summary, the manuscript in its present state reflect an incomplete review rather than providing new insights into the NH ice sheet evolution the last 2.6 Ma. As the data is presented now it is difficult to see what new insights have been gained by this compilation. The asynchronous development of NH ice sheets has been documented previously for the last glacial cycle and the new compilation might show the same pattern for older glaciations although the data density is probably too low to conclude that. This is even acknowledged in the authors (line 115-119).

We envisage this dataset to be of use to the wider Quaternary community, as recognised by Reviewer 1 (‘...provides input for a large scientific community. Several disciplines such as biogeography, paleoclimatology, paleoceanography and paleontology will benefit from this research’) and Reviewer 2 (...’will be of great value to many fields of research, including paleoclimatology, paleoglaciology, numerical modelling of ice sheets and climate, archaeology, and paleontology’).

We do not claim to show asynchronous development of ice sheets for older glaciations, simply that there were variations in the ice-marginal position between each glacial cycle. We have amended the text to make this clear.

Accordingly, I find the manuscript immature for publication. However, it has big potential and I urge the authors to 1) provide a more complete database, and 2) re-assess the way ice margin extents are produced and used in discussing NH ice sheet evolution during the Quaternary.

1)

- We have conducted an additional literature search that has resulted in the inclusion of a further 9 empirically derived ice-sheet outlines and 24 data points.
- We have also added a paragraph to the main text that explains how data were selected for this compilation.

2)

- We have added a new paragraph to the main text that describes and justifies our approach to reconstructing ice-sheet extents in time-slices where empirical data are limited/ absent.
- We have added individual and overall robustness scores to Fig. 1 to stress the level of certainty in each reconstruction.
- We have added a significant amount of text (around 13 pages) to the Supplementary, which better explains the positions of the ice margins in the various time-slices.

REVIEWERS' COMMENTS:

Reviewer #1 (Remarks to the Author):

The authors have clearly improved the initial version of the manuscript, which I personally found already very good. In its new form the manuscript is clearer, carries a better explanation of the applied methods and, in particular, of the empirical data analysis. Importantly, the authors provide with a honest conclusion on the ice-sheets asymmetry that, based on their approach, can only be confidently observed within the last glacial cycle.

In conclusions, I am satisfied with the current status of the manuscript.

Minor issues:

1. Ref. 39: Simms et al. => missing year (line 424-425)

Reviewer #2 (Remarks to the Author):

I have read all the reviewers' comments and the response letter from the authors, and based on the actions they claim to have done, I would gladly recommend publication of this manuscript in Nature Communications.

Reviewer #3 (Remarks to the Author):

Reviewer 3

I am pleased to see the revised version of the manuscript. The authors have considered all the concerns raised by all 3 reviewers. The MS is now well-written, clearly structured and has sound and transparent conclusions supported by data. I have no further comments.

Reviewer #1 (Remarks to the Author):

The authors have clearly improved the initial version of the manuscript, which I personally found already very good. In its new form the manuscript is clearer, carries a better explanation of the applied methods and, in particular, of the empirical data analysis. Importantly, the authors provide with a honest conclusion on the ice-sheets asymmetry that, based on their approach, can only be confidently observed within the last glacial cycle.

In conclusions, I am satisfied with the current status of the manuscript.

Minor issues:

1. Ref. 39: Simms et al. => missing year (line 424-425)

Amended.

We thank the reviewer for noticing this typo and for their helpful and encouraging comments during the initial review stage.

Reviewer #2 (Remarks to the Author):

I have read all the reviewers' comments and the response letter from the authors, and based on the actions they claim to have done, I would gladly recommend publication of this manuscript in Nature Communications.

We thank the reviewer for their positive and encouraging review of this manuscript.

Reviewer #3 (Remarks to the Author):

Reviewer 3

I am pleased to see the revised version of the manuscript. The authors have considered all the concerns raised by all 3 reviewers. The MS is now well-written, clearly structured and has sound and transparent conclusions supported by data. I have no further comments.

We thank the reviewer for their helpful comments on this manuscript.